# Multifactorial processes underlie parallel opsin loss in neotropical bats

Alexa Sadier[1†], Kalina TJ Davies[2†], Laurel R Yohe[3,4‡], Kun Yun[5], Paul Donat[3], Brandon P Hedrick[6], Elizabeth R Dumont[7], Liliana M Dávalos[3,8]*, Stephen J Rossiter[2]*, Karen E Sears[1]*

[1]Department of Ecology and Evolutionary Biology, University of California, Los Angeles, United States; [2]School of Biological and Chemical Sciences, Queen Mary University of London, London, United Kingdom; [3]Department of Ecology and Evolution, Stony Brook University, New York, United States; [4]Geology & Geophysics, Yale University, New Haven, United States; [5]Department of Animal Biology, University of Illinois, Urbana, United States; [6]Department of Organismic and Evolutionary Biology, Harvard University, Cambridge, United States; [7]School of Natural Sciences, University of California, Merced, United States; [8]Consortium for Inter-Disciplinary Environmental Research, School of Marine and Atmospheric Sciences, Stony Brook University, New York, United States

**\*For correspondence:**
liliana.davalos@stonybrook.edu (LMD);
s.j.rossiter@qmul.ac.uk (SJR);
ksears@ucla.edu (KES)

[†]These authors contributed equally to this work

**Present address:** [‡]Department of Geology & Geophysics, Yale University, New Haven, United States

**Abstract** The loss of previously adaptive traits is typically linked to relaxation in selection, yet the molecular steps leading to such repeated losses are rarely known. Molecular studies of loss have tended to focus on gene sequences alone, but overlooking other aspects of protein expression might underestimate phenotypic diversity. Insights based almost solely on opsin gene evolution, for instance, have made mammalian color vision a textbook example of phenotypic loss. We address this gap by investigating retention and loss of opsin genes, transcripts, and proteins across ecologically diverse noctilionoid bats. We find multiple, independent losses of short-wave-sensitive opsins. Mismatches between putatively functional DNA sequences, mRNA transcripts, and proteins implicate transcriptional and post-transcriptional processes in the ongoing loss of S-opsins in some noctilionoid bats. Our results provide a snapshot of evolution in progress during phenotypic trait loss, and suggest vertebrate visual phenotypes cannot always be predicted from genotypes alone.
DOI: https://doi.org/10.7554/eLife.37412.001

## Introduction

The reduction and eventual loss of previously adaptive traits can be seen across the Tree of Life, and is typically linked to relaxation in selection. Within vertebrates, examples of losses include flight in birds, armor plates in sticklebacks, and the ability to synthesize vitamin C in bats (*Burga et al., 2017*; *Cui et al., 2011*; *Le Rouzic et al., 2011*). Strikingly, many instances of trait loss occur in parallel across multiple independent lineages (e.g. *Colosimo et al., 2004*, *Drouin et al., 2011*, and *Harshman et al., 2008*). There have been attempts to relate parallel trait losses to shared ecological conditions such as salinity tolerance or switches in diet, but the precise causal links are not always clear (*Marchinko and Schluter, 2007*). In contrast, the genetic bases of parallel trait loss are often known, with pseudogenization – whereby a non-essential gene loses some functionality – being a frequently invoked mechanism (e.g. *Cui et al., 2011* and *Protas et al., 2006*).

One of the best known examples of parallel phenotypic loss via pseudogenization, which can often be directly related to shifts in ecology, is that of color vision in vertebrates. Opsins encode the

**eLife digest** Bats are famous for using their hearing to explore their environments, yet fewer people are aware that these flying mammals have both good night and daylight vision. Some bats can even see in color thanks to two light-sensitive proteins at the back of their eyes: S-opsin which detects blue and ultraviolet light and L-opsin which detects green and red light. Many species of bat, however, are missing one of these proteins and cannot distinguish any colors; in other words, they are completely color-blind.

Some bat species found in Central and South America have independently lost their ability to see blue-ultraviolet light and have thus also lost their color vision. These bats have diverse diets – ranging from insects to fruits and even blood – and being able to distinguish color may offer an advantage in many of their activities, including hunting or foraging. The vision genes in these bats, therefore, give scientists an opportunity to explore how a seemingly important trait can be lost at the molecular level.

Sadier, Davies et al. now report that S-opsin has been lost more than a dozen times during the evolutionary history of these Central and South American bats. The analysis used samples from 55 species, including animals caught from the wild and specimens from museums. As with other proteins, the instructions encoded in the gene sequence for S opsin need to be copied into a molecule of RNA before they can be translated into protein. As expected, S-opsin was lost several times because of changes in the gene sequence that disrupted the formation of the protein. However, at several points in these bats' evolutionary history, additional changes have taken place that affected the production of the RNA or the protein, without an obvious change to the gene itself. This finding suggests that other studies that rely purely on DNA to understand evolution may underestimate how often traits may be lost. By capturing 'evolution in action', these results also provide a more complete picture of the molecular targets of evolution in a diverse set of bats.
DOI: https://doi.org/10.7554/eLife.37412.002

photoreceptor proteins of rod cells that are responsible for dim-light and cone cells responsible for color vision. Most mammals possess three visual opsins: rhodopsin (RHO) in rods, and opsin one long-wave sensitive (OPN1LW) found in L-cones, and opsin one short-wave sensitive (OPN1SW) found in S-cones. Reconstructions of the highly complex evolutionary history of mammalian vision suggest that there have been >20 independent losses of cone-opsins, with associated reduction in color sensitivity (e.g. *Bowmaker, 1998*, *Emerling et al., 2015*, *Lucas et al., 2003*, *Porter et al., 2012* and *Yokoyama et al., 2008*). This is exemplified in some cetacean and xenarthran lineages, which appear to have lost both of their cone-opsins (*Emerling and Springer, 2015*; *Meredith et al., 2013*).

Evolutionary reconstructions of color vision have nearly all been based solely on opsin gene sequences, with gene expression and protein data limited or missing for most mammalian species, including cetaceans and primates (but see *Kraus et al., 2014*, *Peichl et al., 2017*, *Schweikert et al., 2016* and *Wikler and Rakic, 1990*). To date, no large-scale comparative study of color vision in mammals has considered each of the steps in protein production (e.g., transcription, translation). Thus, the extent to which visual phenotypes are expressed or masked due to the modulation of protein production is currently unknown, raising the possibility of underestimating the true complexity of the evolutionary history of vertebrate color vision. This represents a major gap in our understanding of visual evolution, as mounting evidence from a range of systems reveals that complex post-transcriptional and post-translational routes shape phenotypic variation and complicate genotype-to-phenotype mapping (*Blount et al., 2012*; *Csárdi et al., 2015*; *Schwanhäusser et al., 2011*). Such incomplete information might also lead to erroneous conclusions surrounding the adaptive significance of particular genotypes.

The potential for selection to act on phenotypes at different stages of protein production may be particularly important during rapid functional trait diversification, as is often the case in visual systems. In sticklebacks, for example, the repeated colonization of lakes with different photopic environments has driven shifts in spectral sensitivity via recurrent selective sweeps in short-wave opsin genes, and changes in opsin expression (*Marques et al., 2017*; *Rennison et al., 2016*). Similarly,

rapid shifts in the visual ecology of cichlid fishes have involved a combination of coding sequence evolution and changes in expression (*O'Quin et al., 2010*; *Spady et al., 2005*). However, in contrast to fishes, much less is known about the changes underpinning rapid visual adaptations in mammals and reptiles, for which relevant studies have tended to focus on ancient transitions to nocturnal, aquatic or subterranean niches (*Emerling, 2017*; *Emerling et al., 2017*; *Jacobs et al., 1993*).

In this study, we investigate the molecular signatures of the repeated loss of S-opsins, and associated dichromatic and UV-vision capabilities, in bats of the superfamily Noctilionoidea (~200 species of New World leaf-nosed bats and allies within the suborder Yangochiroptera). These bats underwent ecological diversification approximately 40 million years ago (*Rojas et al., 2012*; *Rossoni et al., 2017*), and show marked morphological and sensory adaptations linked to their unparalleled dietary specializations (*Davies et al., 2013*; *Dumont et al., 2012*; *Hayden et al., 2014*; *Monteiro and Nogueira, 2010*; *Yohe et al., 2017*). Switches in feeding ecology from generalized insectivory to blood-, insect-, vertebrate-, nectar- or fruit-based diets have occurred multiple times among closely related species, making noctilionoid bats an outstanding group in which to examine the genetic basis of visual adaptations.

Until recently it was thought that S-opsin, encoded by the *OPN1SW* gene, was likely functional across the suborder Yangochiroptera (e.g. *Butz et al., 2015*, *Feller et al., 2009*, *Marcos Gorresen et al., 2015*, *Gutierrez et al., 2018*, *Müller et al., 2009*, *Winter et al., 2003* and *Zhao et al., 2009a*). However, with increased taxonomic sampling of neotropical bat species this has been shown to not be the case, and multiple independent lineages with diverse ecologies (e.g. blood feeding, plant-visiting species) show evidence of *OPN1SW* pseudogenization (*Kries et al., 2018*; *Li et al., 2018*; *Wu et al., 2018*). Notably, lineages shown to have lost their S-opsins – and thus by association UV-sensitivity – are from the Noctilionoidea superfamily. In contrast, within the other bat suborder – the Yinpterochiroptera – multiple losses of S-opsin function have previously been documented in lineages of Old World fruit bats as well as horseshoe bats and Old World leaf-nosed bats that have evolved a derived form of laryngeal echolocation (*Zhao et al., 2009a*). The loss of S-opsins could have profound impacts on bat visual acuity, as inferences from amino acid sequence analyses and action spectra suggest that bat short-wave opsins are sensitive to UV, and their retention is possibly related to the demands of visual processing in mesopic, or low-light, conditions (*Zhao et al., 2009a*), and/or plant visiting (*Butz et al., 2015*; *Feller et al., 2009*; *Kim et al., 2008*; *Müller et al., 2009*; *Müller et al., 2007*). However, the limited taxonomic sampling to date has precluded clear conclusions. Similarly, while the taxonomic sampling of the recent studies of neotropical bat vision (e.g. *Gutierrez et al., 2018*, *Kries et al., 2018*, *Li et al., 2018*, and *Wu et al., 2018*) is considerably more extensive than previous work, it remains limited, and the functionality of *OPN1SW* has been based on analyses of DNA sequences and a few transcriptome samples.

To determine whether patterns of adaptations and loss in cone opsins (OPN1SW and OPN1LW) in noctilionoid bats are associated with ecological factors such as diet shifts, we applied analyses of sequence evolution, gene expression, and immunohistochemistry across the taxonomic and ecological breadth of the clade and outgroup taxa. For the first time in mammals, our findings reveal that extensive losses of S-opsin gene function can result from disruption at all three levels of protein synthesis (i.e. DNA open-reading frame, mRNA, and protein). Furthermore, we identify three putative molecular routes that may lead to disruptions of protein synthesis leading to the loss of S-opsins in key lineages. In each instance, the specific route to loss of function was seen in multiple independent lineages. Thus, across the noctilionoids we find evidence that parallel losses leading to identical phenotypes have arisen by both the same and different failures of translation. Hence, current studies both underestimate the extent of parallel losses, and might lead to an incomplete picture based on genes alone.

## Results

Our comprehensive analyses of visual opsins that combined information from DNA, mRNA transcripts, and proteins across noctilionoid bats revealed unexpected variation, as well as evidence of extensive parallel losses in S-opsins that have arisen from failures at multiple stages of protein synthesis. First, we used immunohistochemistry to characterize and quantify S- and L-opsin proteins in the retinas of adult bats. Second, for a subset of these taxa, we performed RNA-Seq to assess the presence or absence of transcripts for *OPN1SW*, *OPN1LW*, and *RHO*, and estimated the mode and

the strength of selection in coding sequences. Finally, we modeled the presence or absence of S-opsin intact ORF, mRNA or protein presence, as a function of dietary and roosting ecology.

## Pervasive parallel losses of shortwave opsin pigments in neotropical bats

To assess the presence or absence of OPN1SW and OPN1LW proteins we applied immunohistochemistry (IHC) to whole, flat-mounted retinas of adult bats ($n_{eyes}$ = 218, $n_{individuals}$ = 187, $n_{species}$ = 56). While the presence of a given protein does not guarantee its functionality, here we interpret the detection of protein as indicating a functional cone in the absence of contradictory evidence. Since the absence of protein is difficult to assess, we applied quality control (see Materials and methods for the criteria to accept or reject a retina based on its condition or number of replicates). Surprisingly, OPN1SW was only detected in just over half of the species assayed ($n$ = 32), including all members of the primarily frugivorous subfamily Stenodermatinae, which invariably retain their S-cones (*Figure 1* and *Figure 1—figure supplements 1* and *2*). In contrast, OPN1SW was found to be absent in approximately one third of species assayed ($n$ = 18), including representative species from five bat families. Thus, we not only find evidence of widespread loss of S-opsins within the noctilionoids but also find the first evidence of S-opsin loss in non-noctilionoid Yangochiroptera (*Chilonatalus micropus*, *Eptesicus fuscus* and *Molossus molossus*).

To test whether a lack of signal corresponds to a loss of opsin, we aligned the opsin gene sequences among species and confirmed that the epitope-binding site was relatively conserved and showed no correspondence with loss (*Figure 1—figure supplement 3*, *Figure 1—figure supplement 4*). For immunohistochemistry, five bat species had replicates that were both wild-caught and from museum collections and exhibited the same phenotype, highlighting the robustness of the experiments (*Figure 1—figure supplement 5*). We also verified absences using multiple replicates of slides from the same individuals, and, when available, from multiple individuals from the same species (with a minimum of two individuals; see *Supplementary file 1*). In one species, *P. quadridens,* we detected evidence of polymorphism in the presence of S-opsin cones among fresh specimens, with three of the 17 specimens lacking S-cones. Samples derived from museum specimens of six species had low signal-to-background ratios in OPN1SW protein labeling (e.g. opsin-specific staining was seemingly detected, but non-specific background staining was high, making specific staining difficult to distinguish from background), generating inconclusive results, and these specimens were therefore excluded from our models. In contrast to our OPN1SW results, we detected OPN1LW protein in all species examined (*Figure 1* and *Figure 1—figure supplements 1* and *2*).

We then analyzed patterns of OPN1SW and OPN1LW protein localization among cells. Consistent with cone-specific roles, we found that almost all cones expressed either OPN1SW or OPN1LW protein, with no strong evidence of co-localization of both proteins (*Figure 2* and *Figure 1—figure supplements 1* and *2*).

## Multiple parallel losses in transcripts across taxa

To investigate the underlying molecular causes of the above-detected losses of OPN1SW, we began by sequencing total mRNA isolated from the eye tissue of 39 species, assembled the short-read data and used a BLAST approach to annotate visual pigments. We found evidence of at least partial *OPN1SW* mRNA transcripts in a total of 34 bat species; thus, expression was only absent in five of the species assayed. Absences of *OPN1SW* transcripts were phylogenetically widespread, and included divergent species from three families (Natalidae: *Chilonatalus micropus*; Mormoopidae: *Mormoops blainvillei* and Phyllostomidae: *Macrotus waterhousii*, *Brachyphylla (nana) pumila* and *Lionycteris spurrelli*) (see *Figure 1* and *Supplementary file 1*). In addition to losses observed in *C. micropus* and *M. blainvillei*, the three phyllostomid species each belong to separate subfamilies, and therefore, likely represent independent losses of *OPN1SW* expression. In comparison, we were able to recover the complete *RHO* and *OPN1LW* transcript from all taxa assessed ($n$ = 39; *Figure 1*).

## Mismatches in opsin transcript and protein suggest alternate parallel failures of translation

While IHC revealed pervasive loss of S-cones across our study sample, we only detected loss of the *OPN1SW* transcript in a few species. Comparison of our *OPN1SW* transcript and protein data

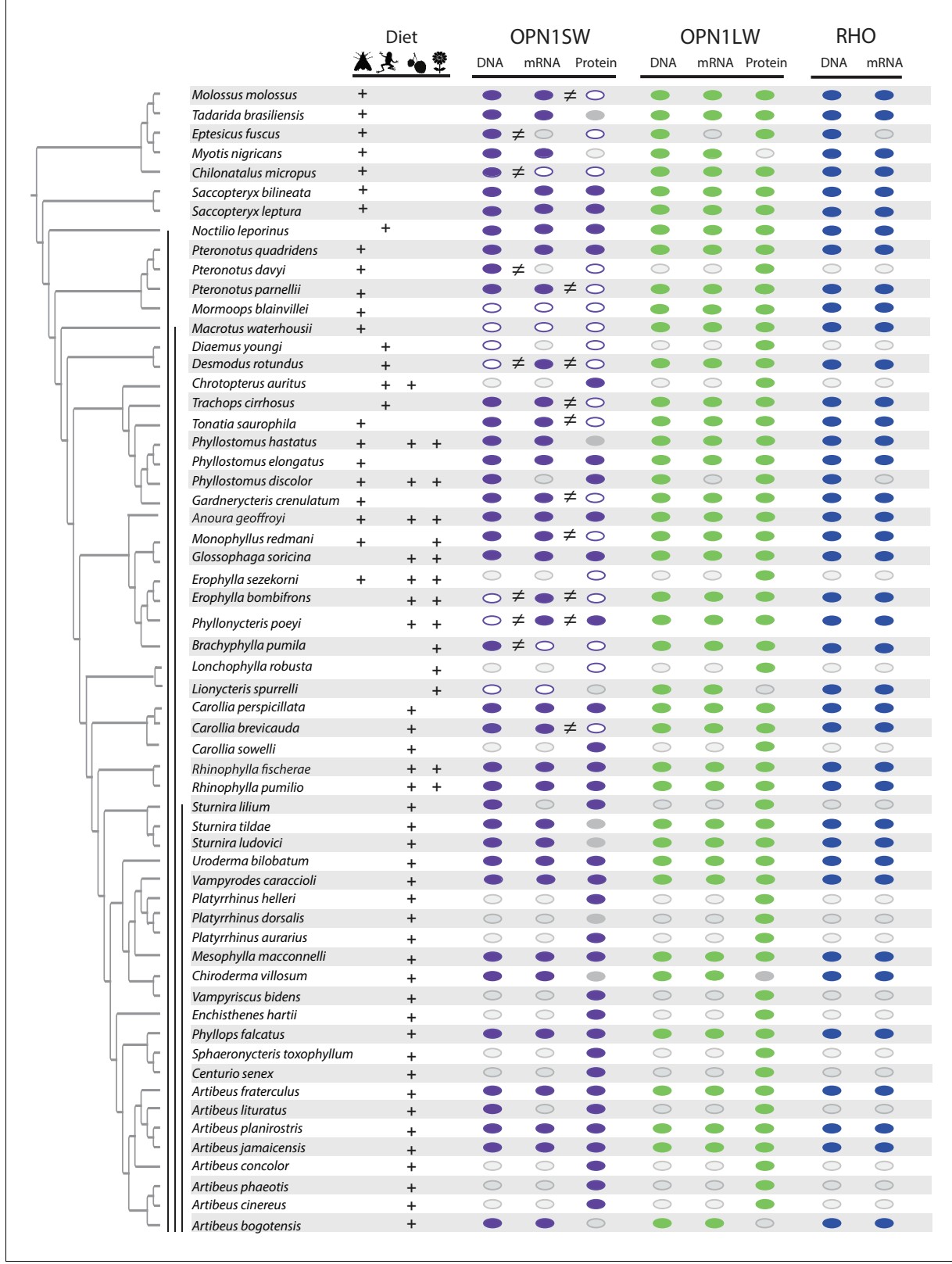

**Figure 1.** Distribution of an intact open reading frame (ORF), mRNA transcript, and protein for the OPN1SW, OPN1LW, and RHO photopigments in ecologically diverse noctilionoid bats. The composition of species diet follows *Rojas et al. (2018)*, dietary types are indicated with the following symbols: invertebrates – moth, vertebrates – frog, fruit – fruit and nectar/pollen – flower. The species phylogeny follows *Rojas et al. (2016)* and *Shi and Rabosky (2015)*. Vertical black bars, from left to right, indicate: (1) Noctilionoidea, (2) Phyllostomidae, (3) and Stenodermatinae. RNA-Seq data was

*Figure 1 continued*

generated to both infer the presence of an intact ORF (in combination with genomic and PCR sequence data) and to determine the presence of an expressed mRNA transcript. The presence of an intact ORF and mRNA transcript for RHO was verified across all transcriptomes. The presence/absence of a protein product for S- and L-opsins was assayed by IHC on flat mounted retinas. The presence of an intact ORF, mRNA, and protein are indicated by a filled color marker (*OPN1SW* – purple, *OPN1LW* – green and *RHO* – blue), and its absence by a white marker. Missing data (i.e. species for which we were unable to obtain tissue) are indicated with a grey marker with grey outline. Mismatches between intact ORFs and transcripts, or between transcripts and protein data are indicated by an inequality symbol. Note: OPN1SW protein assays for *P. quadridens* revealed polymorphisms within the sample, and we recorded positive OPN1SW assays in some *P. poeyi* individuals despite an apparent disrupted ORF. Finally, a grey marker with no outline indicates the failure of protein assay for some species represented by museum specimens (*Tadarida brasiliensis*, *Phyllostomus hastatus*, *Sturnira tildae*, *Sturnira ludovici*, *Platyrrhinus dorsalis* and *Chiroderma villosum*).

DOI: https://doi.org/10.7554/eLife.37412.003

The following figure supplements are available for figure 1:

**Figure supplement 1.** L- and S-opsin protein expression in the L- and S-cones of field samples

DOI: https://doi.org/10.7554/eLife.37412.004

**Figure supplement 2.** L- and S-opsin protein expression in the L- and S-cones of museum samples.

DOI: https://doi.org/10.7554/eLife.37412.005

**Figure supplement 3.** Partial amino acid alignments for OPN1SW across the bat species studied.

DOI: https://doi.org/10.7554/eLife.37412.006

**Figure supplement 4.** Partial amino acid alignments for OPN1LW across the bat species studied.

DOI: https://doi.org/10.7554/eLife.37412.007

**Figure supplement 5.** L- and S-opsin protein expression in L- and S-cones visualized in museum specimen of various ages (from 1921 to 1998) showing the consistency of the staining in old specimens.

DOI: https://doi.org/10.7554/eLife.37412.008

revealed numerous conflicts in species-specific absences, with a total of nine lineages found to possess *OPN1SW* transcripts but lack OPN1SW protein (see *Figures 1*, *3* and *4*). These species included *Molossus molossus*, *Pteronotus parnellii*, *Desmodus rotundus*, *Trachops cirrhosus*, *Tonatia saurophila*, *Gardnerycteris crenulatum*, *Monophyllus redmani*, *Erophylla bombifrons*, and *Carollia brevicauda*. This may also be the case in additional species, for example *Tadarida brasiliensis*, *Eptesicus fuscus*, *Pteronotus davyi* and *Diaemus youngi*, but we currently lack the complementary data to confirm this.

Of the nine species lacking S-cones, but in which the presence of mRNA transcripts was detected, we further examined the nucleotide sequence of the assembled transcripts for both an intact open-reading frame (ORF) and completeness of transcript. Only a single partial mRNA fragment was recovered each for *D. rotundus* and *M. redmani*. The partial *D. rotundus* fragment (~300 base-pairs of exons 2–4) contained a premature stop codon – confirmed by PCR and the recently published common vampire bat genome. The partial *M. redmani* fragment (~240 base-pairs of exons 2–3) did not contain premature stop codons or indels; however, we cannot rule out the possibility that these may be present in the remaining exons not sequenced. We recovered a total of three *OPN1SW* transcripts for *E. bombifrons* from our transcriptome assembly, and the longest of these transcripts contained a putative four base-pair deletion and retained a portion of the intron between exons 2 and 3. Therefore, *D. rotundus* and *E. bombifrons* appear to have transcribed *OPN1SW* pseudogenes.

For the remaining six species (*M. molossus*, *P. parnellii*, *T. cirrhosus*, *T. saurophila*, *G. crenulatum* and *C. brevicauda*) for which the *OPN1SW* transcript was present but the S-cone protein was absent several alternative scenarios emerged. First, for two species, *C. brevicauda* and *T. saurophila*, our RNA-Seq assemblies recovered a single and complete *OPN1SW* mRNA transcript containing all five exons and the 3' UTR, albeit with ~5 codons missing at the 5' end in one of these taxa (*C. brevicauda*). The individuals sequenced for *M. molossus* revealed five *OPN1SW* transcript isoforms, two for *T. cirrhosus* and four for *G. crenulatum*. While the complete transcript (i.e. exons 1–5, and no intronic sequences) was detected in each of these species, we also found evidence of alternative splice variants characterized by either missing exons (*M. molossus*), or retained introns. As a result, the reason for the apparent failure of the S-opsin translation is unclear. Finally, for *Pteronotus parnellii* (n = 4), we detected the most splice variation, with 2, 4, 5, or 18 variants assembled per individual. Furthermore, in *P. parnellii*, we were unable to recover any intact mRNA isoforms among these variants. Many of the variants were repeated across individuals; for example, an entirely missing exon

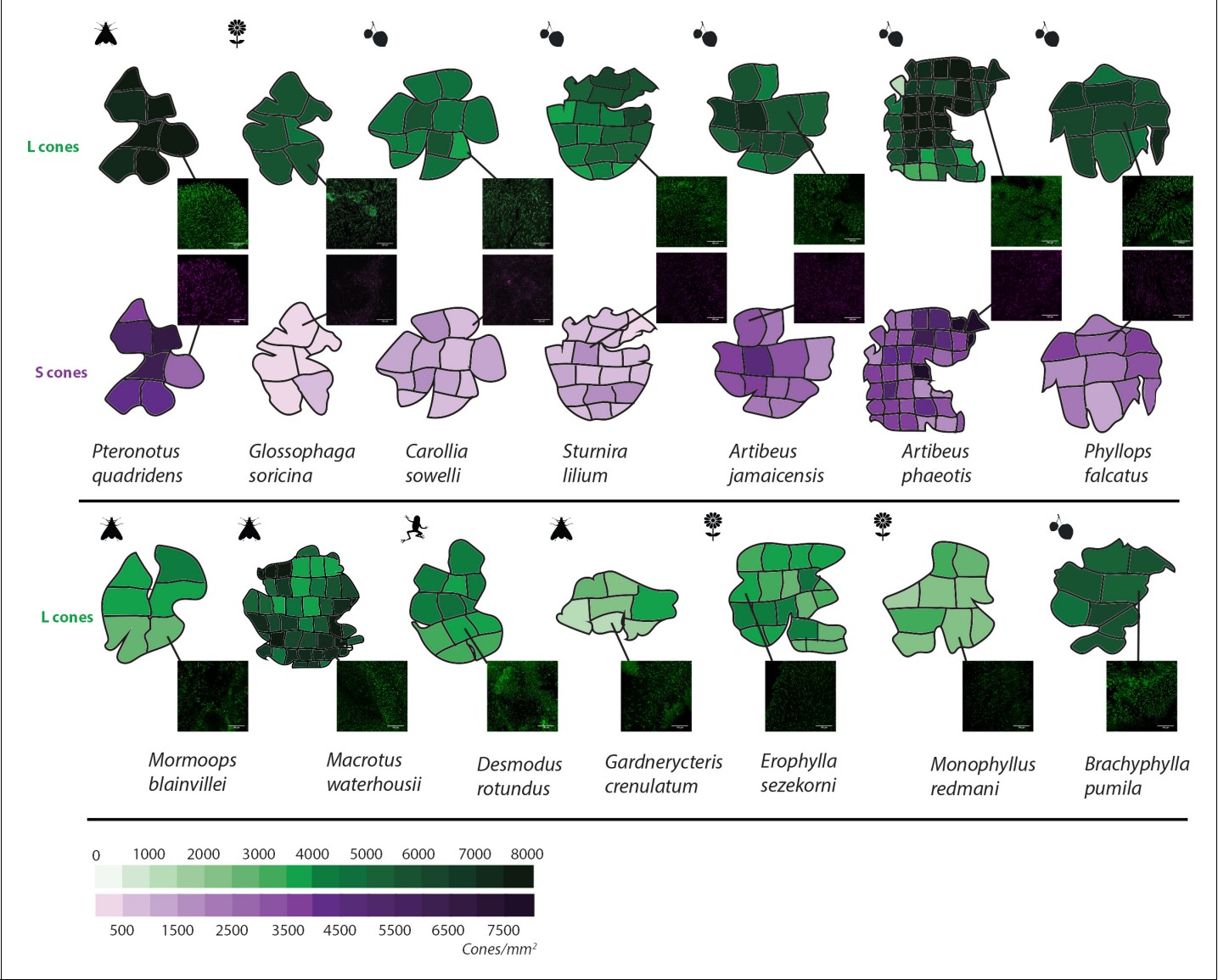

**Figure 2.** L and S opsin cone distribution in 14 representative noctilionoid bat species. Density maps of L and S opsin cone topography in 14 noctilionoid bat species. For each species, a representative dissected flat-mounted retina is shown. Insets are representative IHC magnifications of flat mounted retinas immune-stained for either L- or S-opsins in the highlighted region. Dietary types are indicated with the following symbols: invertebrates – moth, vertebrates – frog, fruit – fruit and nectar/pollen – flower. Measured opsin densities (0–8000 cones/mm²) are represented by the following color scales: L-opsins – green and S-opsins – purple.

DOI: https://doi.org/10.7554/eLife.37412.009

five (despite the presence of the up-/down-stream sequences), and partial deletion of exon 1, was seen in two individuals (see *Figure 4*). We thus speculate that these splice variants explain the observed failure of translation that leads to the absence of OPN1SW protein in *P. parnellii*. We also detected an in-frame three base-pair deletion (Y190del) in three of four *P. parnellii* individuals sequenced. A similar pattern of isoform variants, as detected in *P. parnellii*, was also seen in *P. quadridens* (*n* = 1). Therefore, given the polymorphic status of S-cones in *P. quadridens*, we speculate that the individual sequenced may not have had functional S-cones.

The retention of intronic sequence, while unexpected, does not necessarily indicate a non-functional gene by itself. For example, we detected limited evidence of some intron retention in species for which the protein data suggest S-opsin presence, for example, *Artibeus jamaicensis* and *Phyllops falcatus*. Furthermore, across the 39 species, we found some evidence of *OPN1LW* transcript

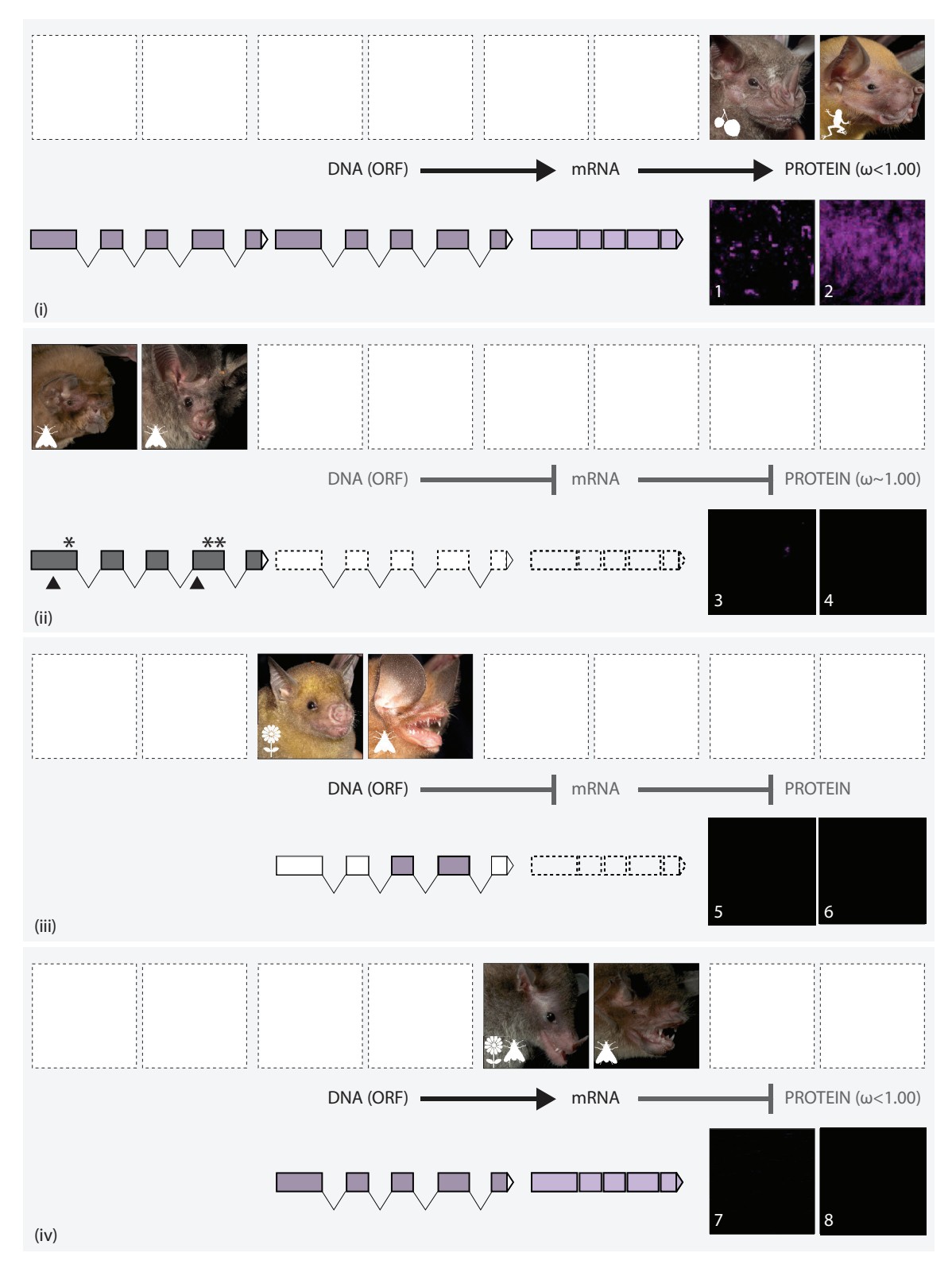

**Figure 3.** The putative routes explaining variation in S-cone presence in noctilionoid bats. In each panel, upper images (left and right) show the gross phenotype of the eye in representative bat species, and lower images (numbered) show IHC magnifications of their respective flat mounted retinas immune-stained for S-opsin. Diets are depicted with the following symbols: invertebrates – moth, vertebrates – frog, fruit – fruit and nectar/pollen – flower. (i): Information in the intact DNA Open Reading Frame (dark purple) is transcribed to form mRNA (light purple), which is then translated into

*Figure 3 continued on next page*

*Figure 3 continued*

OPN1SW. Codon analyses reveal purifying selection. Example species: (L + 1) *Artibeus jamaicensis*; (R + 2) *Noctilio leporinus*. (ii): The DNA ORF is disrupted (grey) by the presence of STOP codons (*) and indels (black triangles). Neither *OPN1SW* mRNA (dashed boxes) nor OPN1SW are detected. Codon analyses reveal relaxed selection. Example species: (L + 3) *Mormoops blainvillei*; (R + 4) *Macrotus waterhousii*. (iii): Although the DNA ORF (dark purple) appears to be intact, information is not transcribed to mRNA (dashed boxes), and no OPN1SW is detected. Example species: (L + 5) *Brachyphylla pumila* (R + 6) *Chilonatalus micropus*. (iv): Information in the intact DNA ORF (dark purple) is transcribed to form mRNA (light purple); however, the OPN1SW is not detected. Codon analyses reveal purifying selection. Example species: (L + 7) *Monophyllus redmani* (R + 8) *Pteronotus parnellii*..

DOI: https://doi.org/10.7554/eLife.37412.010

variation in 15 individuals (14 species), with instances of retained introns, missing exons and, in one case (*Artibeus planirostris*) an indel, although this latter case may arise from assembly error. For RHO, we also found some evidence of retained introns for three species (*Anoura geoffroyi*, *T. brasiliensis* and *T. cirrhosus*). However, summing across these 16 species, and except for *T. cirrhosus*, we always recovered an isoform with all exons and no intronic sequences for both *RHO* and *OPN1LW*.

Among the four species lacking the OPN1SW transcript and protein, we found the ORF recovered by manual PCR of exons 3–4 was intact in *C. micropus* (the single sampled Natalidae species), as well as the unrelated phyllostomid *B. (nana) pumila*. In contrast, the genomic sequences recovered by blastn was disrupted by a number of insertions and deletions resulting in premature stop codons in *Mormoops blainvillei*, *Macrotus waterhousii*, and *Lionycteris spurrelli*.

In the 17 species for which the S-opsin protein was detected and for which we were also able to test the presence of the transcript, we found a 1-to-1 correspondence in all but one case. The exception was *Phyllonycteris poeyi*, for which data from four individuals showed S-cone presence, yet the transcript of one other individual was inferred to be non-functional based on a four base-pair insertion (confirmed via PCR). We also note that within the *OPN1SW* sequences of most species examined, the ATG start codon was found to be three codons downstream relative to that of the human orthologue (*Figure 1—figure supplement 2*). In contrast to the mismatches between *OPN1SW* mRNA and OPN1SW protein, we found complete correlation between the presence of the transcript and protein for OPN1LW in these species.

## Molecular evolution of opsin genes

To gain further insights into the molecular evolution of opsins, we performed tests of divergent selection in alignments for each of the three opsin genes (*OPN1SW*, *OPN1LW* and *RHO*) among three types of lineages: (1) those with S-cones; (2) those without S-cones but with *OPN1SW* transcripts and an intact *OPN1SW* ORF; and (3) those without either S-cones or *OPN1SW* transcripts (see *Figure 5—figure supplement 1*). We found a significantly higher $\omega$ in lineages with a pseudogenized *OPN1SW* in both the *OPN1SW* gene ($\omega_{background}$ = 0.13; $\omega_{OPN1SW.intact}$=0.24; $\omega_{OPN1SW.pseudo}$=0.78; $\chi^2_{(2)}$=70.99, p=3.84e-16), and the *OPN1LW* gene ($\omega_{background}$ = 0.08; $\omega_{OPN1SW1.intact}$=0.08; $\omega_{OPN1SW.pseudo}$=0.19; $\chi^2_{(2)}$=9.18, p=0.01). In contrast, we found no differences in $\omega$ across the different lineages for *RHO*, indicating strong and negative selection in this gene (Table S2 in *Supplementary file 2*). To test the influence of diet on rates of molecular evolution, we compared $\omega$ for all three opsin genes between frugivorous and non-frugivorous lineages. We found no differences in rates for *OPN1SW* or *OPN1LW*, however background branches (non-frugivorous) had a significant and slightly higher $\omega$ for *RHO* ($\omega_{background}$ = 0.04; $\omega_{fugivory}$= 0.01; $\chi^2_{(1)}$=13.77, p=2.07e-4; Table S3 in *Supplementary file 2*).

## Ecological correlates of opsin presence and density

We compared the locations and densities of long-wavelength-sensitive cones (or L-cones expressing OPN1LW protein) and short-wavelength-sensitive (or S-cones expressing OPN1SW protein) in the whole, flat-mounted retinas of adult bats for 14 species for which we had sufficient specimen replicates (Table S4 in *Supplementary file 2*). We found photoreceptor density varied among examined species (*Figure 2*, Table S4 in *Supplementary file 2*), with mean cone densities ranging from 2500 to 7500 cones/mm$^2$ for L-cones, and 327 to 5747 cones/mm$^2$ for S-cones (*Figure 2*, Table S4 in *Supplementary file 2*). In every case, the density of S-cones was lower than that of L-cones. Densities of both cone types tended to be highest near the center of the retina in all species. Bat

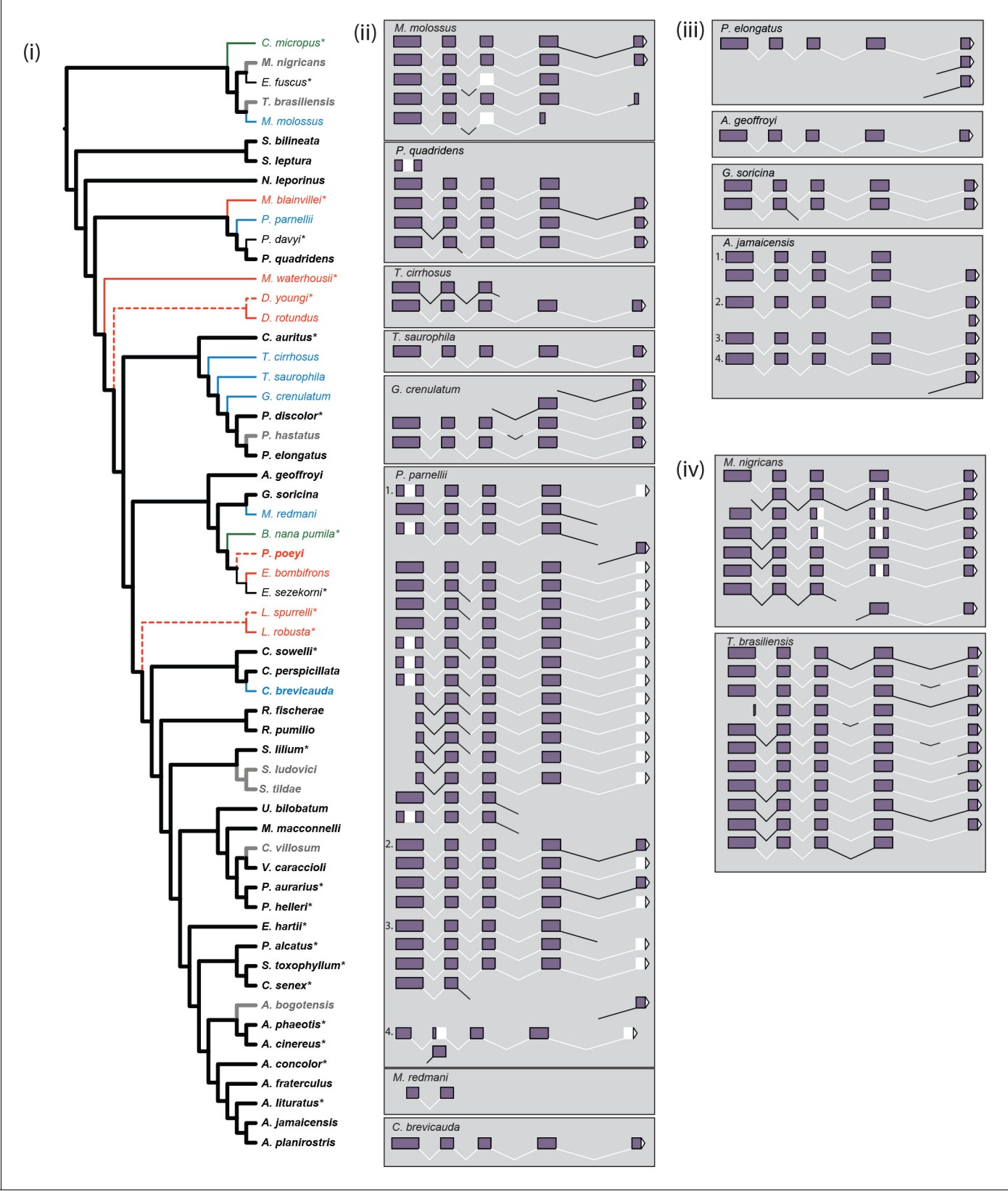

**Figure 4.** Inferred parallel losses of S-opsins mapped on to the species phylogeny and exonic content of reconstructed mRNA. (i) Taxa and branches are colored as follows: presence of protein – black; presence of intact ORF but no protein – green; presence of mRNA but absence of protein – blue; evidence of pseudogenization (disrupted ORF) – red; protein status not determined – grey. Weight of branches indicates: inferred presence of protein
*Figure 4 continued on next page*

*Figure 4 continued*
– heavy; inferred absence of protein – light; protein absence based on evidence of gene loss but not confirmed by IHC – dashed light. We were either not able to recover mRNA, or preserved material was not available, for species marked with '*', evidence for ORF status for *Diaemus youngi* taken from *Kries et al., 2018*. The species phylogeny follows *Rojas et al., 2016* and *Shi and Rabosky, 2015*. (ii) Reconstructed mRNA transcript variants of seven species (*M. molossus, T. cirrhosus, T. saurophila, G. crenulatum, P. parnellii, M. redmani,* and *C. brevicauda*) with OPN1SW mRNA present but no detected protein, and *P. quadridens* for which presence of detected protein varied across individuals. The four biological replicates of *P. parnellii* are numbered 1–4. Sections of the mRNA are indicated as follows: exons 1–5 – purple filled boxes; introns 1–4 – black lines; the 3'UTR – white filled triangle; and missing regions – white regions. (iii) Reconstructed mRNA transcript variants of four species (*P. elongatus, A. geoffroyi, G. soricina* and *A. jamaicensis*) with OPN1SW mRNA present and detected protein. The four biological replicates of *A. jamaicensis* are numbered 1–4. Sections of the mRNA are indicated as above. (iv) Reconstructed mRNA transcript variants of two species (*M. nigricans* and *T. brasiliensis*) with OPN1SW mRNA present but protein status not determined. Sections of the mRNA are indicated as above.
DOI: https://doi.org/10.7554/eLife.37412.011

species with S-cones had significantly higher densities of L-cones, with the presence of S-cones increasing the ln-transformed density of L-cones by 43%, explaining on average ~24% of the variance in density between species (*Table 1*).

We tested the influence of ecology on the presence of the *OPN1SW* ORF, mRNA, or S-cones using Bayesian hierarchical models in which the observations corresponded to species (*OPN1SW* ORF $n_{species} = n_{observations} = 45$, mRNA $n_{species} = 39$, $n_{species} = 50$), and a phylogenetic structure of the data was included as a species-specific effect. Two types of predictor variables were analyzed: three variables for diet and one for roosting. Comparisons of the coefficients, which are all at the same scale because they are multipliers of the presence of a particular ecology, showed frugivory was the best factor for explaining the presence of S-cones (*Figure 5*, Tables S5-S7 in *Supplementary file 2*). The predominance of fruit in the diet increases the odds of having S-cones roughly 39 times, and explains about 50% of the between-species variance in the presence of the S-cone (Table S7 in *Supplementary file 2*, *Figure 5*).

## Discussion

By studying opsin gene sequences, transcripts and proteins across a radiation of neotropical bats (suborder Yangochiroptera, superfamily Noctilionoidea), we discovered remarkable diversity in visual genotypes and phenotypes, including parallel loss of function in *OPN1SW* and the associated S-cones. Although parallel losses have been reported before, the multiple steps involved in functional loss were hitherto unsuspected. We evaluate both the diversity of paths to acquiring a similar phenotype, and the ecological covariates that may explain evolution in this system.

**Table 1.** Summary of Bayesian regression models of the presence of S-cones or the ln-transformed density of L-cones as a function of predictor variables.
$R^2$, variance explained by sample-wide factors; $\Sigma$, species-specific phylogenetic effect for species.

| Formula | $R^2$ | Parameter | Median | 2.5% | 97.5% |
|---|---|---|---|---|---|
| Presence$_i$ ~ a + b. fruit_prevalent$_i$ + $\Sigma$ | 0.503 | intercept (*a*) | −1.10 | −3.19 | 0.52 |
| | | *b* | 3.66 | 1.85 | 6.76 |
| | | $\Sigma$ | 1.94 | 0.02 | 8.64 |
| ln(density)$_i$ ~ a + b. fruit_prevalent$_j$ + $\Sigma_j$ | 0.000 | intercept (*a*) | 8.42 | 8.00 | 8.84 |
| | | *b* | 0.22 | −0.21 | 0.66 |
| | | $\Sigma_j$ | 0.14 | 0.04 | 0.46 |
| ln(density)$_i$ ~ a + b. S-cones_present$_j$ + $\Sigma_j$ | 0.240 | intercept (*a*) | 8.28 | 7.95 | 8.63 |
| | | *b* | 0.43 | 0.09 | 0.78 |
| | | $\Sigma_j$ | 0.07 | 0.000 | 0.51 |

DOI: https://doi.org/10.7554/eLife.37412.012

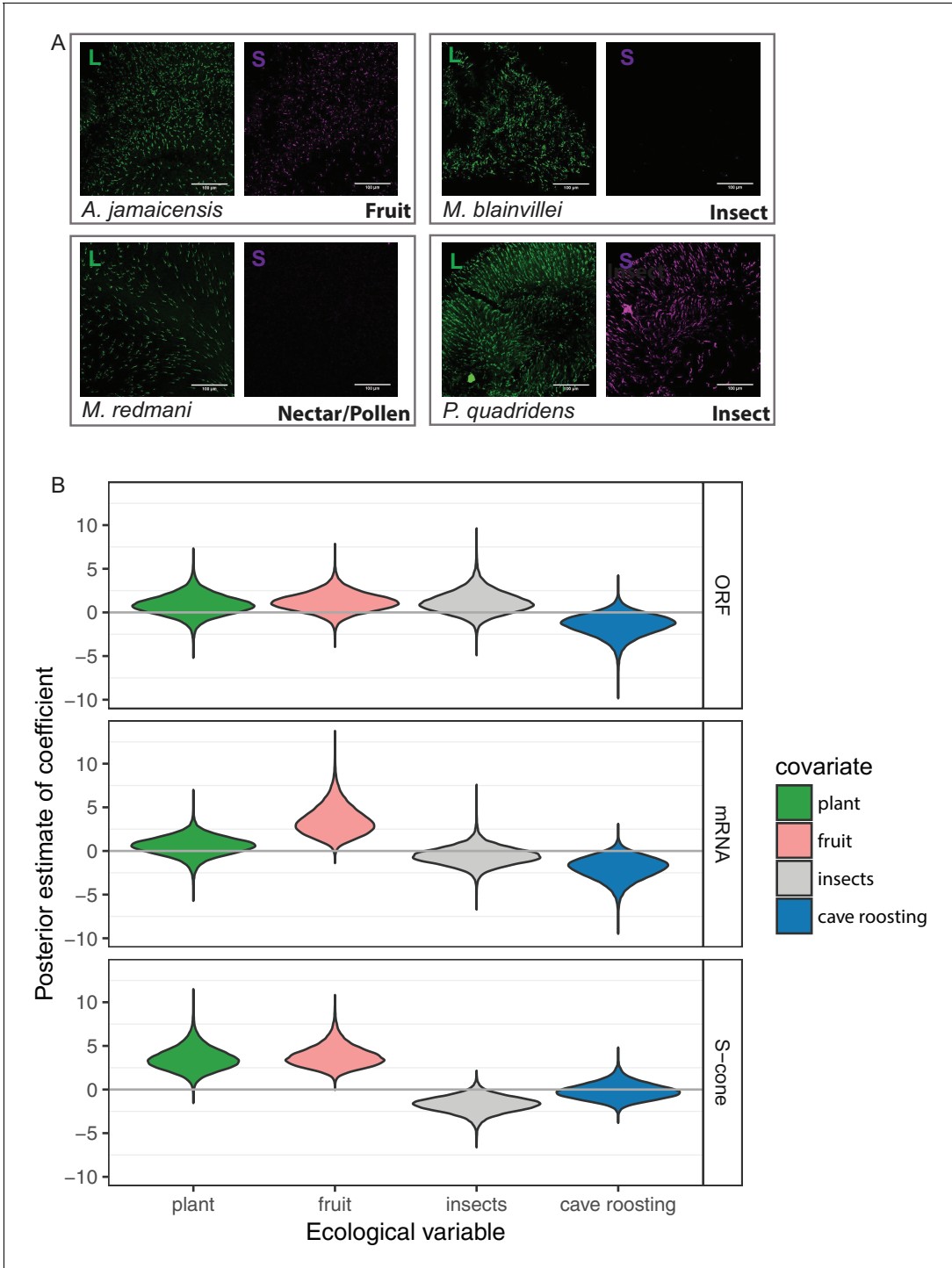

**Figure 5.** S-opsin presence is correlated with diet. (**A**) Representative IHC magnifications of flat mounted retinas immune-stained for both L and S opsin for four species representatives of the diversity of phenotypes observed. Fruit-based diet: *Artibeus jamaicensis*, pollen/nectar-based diet: *Monophyllus redmani*, and insect-based diet: *Mormoops blainvillei* and *Pteronotus quadridens*. (**B**) Violin-plots of the posterior estimates of the coefficients for the presence of an *OPN1SW* ORF, mRNA, or cone as a function of ecological covariates. The gray horizontal lines indicate a coefficient of 0, or no effect of the covariate on the response. The high-probability density estimates for all coefficients are given in Tables S5-S7, and for fruit in *Table 1*.

DOI: https://doi.org/10.7554/eLife.37412.013

The following figure supplement is available for figure 5:

**Figure supplement 1.** Branch class coding for molecular evolution analyses in PAML.

DOI: https://doi.org/10.7554/eLife.37412.014

## Complex independent routes of phenotypic loss

Across our study taxa, we documented molecular signatures consistent with as many as 17 instances of parallel loss in S-opsins, and equated these with a minimum of three putative routes leading to the failure of the formation of the S-opsin cones (*Figures 1*, *3* and *4*). First, we found evidence of multiple independent instances of pseudogenization, associated with either absent or fragmentary mRNA transcripts. Second, we found evidence of an apparently intact gene sequence that did not result in a mRNA transcript. Finally, we recovered putatively intact *OPN1SW* transcripts that did not result in the corresponding S-cone protein, which in some species appears to arise from aberrant isoforms.

The first route of parallel loss in OPN1SW in noctilionoid bats involved disruption in opsin reading frames, a finding that has been documented in other mammals (e.g. *Emerling et al., 2015*, *Emerling et al., 2017*, *Hunt and Peichl, 2014*, *Kraus et al., 2014* and *Zhao et al., 2009a*), including bats from both suborders (e.g. *Emerling et al., 2015*, *Kim et al., 2008*, *Kries et al., 2018*, *Müller et al., 2007*, *Wu et al., 2018* and *Zhao et al., 2009a*). In general, pseudogenization is thought to occur relatively frequently within mammalian genomes, and previous estimates suggest several thousand pseudogenes may be present per genome (e.g. *Torrents et al., 2003*). The recent proliferation in published genomes has also led to increased efficiency in the detection of this form of gene function loss, and typically this is one of the most frequently cited mechanisms of gene loss (e.g. *Emerling et al., 2018* and *Jebb and Hiller, 2018*).

The second inferred form of parallel *OPN1SW* loss in which a putatively intact open reading frame exists but appears not to be transcribed was found in two highly divergent bat species. To our knowledge such mismatches between opsin coding DNA and mRNA have not previously been documented in mammals, including bats, although this largely reflects a shortage of suitable datasets. Indeed, obtaining material with intact mRNA is challenging, and few studies have been able to test the commonly held assumption that an intact ORF equates to functionality. Additional research is thus necessary to confirm the existence, extent and underlying mechanism of ORF-transcript mismatched in bats and other groups. Possible explanations for our observed mismatches include regulatory elements and epigenetic modifications, but a lack of genomic resources for these species precludes more detailed investigations at the present time.

The most widely detected form of parallel loss of S-opsins, seen in six species, was associated with the apparent failure of the expressed *OPN1SW* transcript to be translated into protein. Of these species, protein data for three were obtained from field specimens (<4 years old), two from museum samples, and one from both field (<3 years old) and museum samples. Our inspection of the transcript repertoires of these affected species suggests that mismatches might arise from multiple molecular routes. Across the sampled bats, we found evidence of the expression of multiple mRNA isoforms, that in many cases contained either retained introns or skipped exons, both of which are likely to impede translation. Four individuals of *P. parnellii* exemplify this transcript variation, as none of many mRNA transcripts among these individuals were complete. Reports of read-through of introns, and the skipping of exons, are becoming increasingly common (e.g. *Gaidatzis et al., 2015*, *Wen et al., 2018* and *Wong et al., 2016*), and these have previously been linked to loss of gene function (e.g. *Lopes-Marques et al., 2018*). Indeed it is particularly noteworthy that the *OPN1SW* mRNA of the blind mole-rat *Spalax ehrenbergi* has also been found to contain introns (*David-Gray et al., 2002*; *Esquiva et al., 2016*). Underlying mechanisms for these cases could potentially include mutations leading to loss of splice sites, the evolution of novel cryptic splice sites or a reduction of spliceosome efficiency (*David-Gray et al., 2002*; *O'Neill et al., 1998*). We note, however, that in species other than *P. parnellii* either a single, complete mRNA transcript was recovered, or at least one of the alternative assemblies represented the complete transcript – therefore, the ultimate molecular cause of the failure of the protein to be synthesized is unclear in these cases.

Although the causal mutations or mechanisms underpinning losses of function in *OPN1SW* are currently not known, the observed absence of expression with putatively intact ORFs in some species, alongside the converse condition in *D. rotundus*, strongly indicates independent routes. Similar diversity is seen in the Mormoopidae in which we detected a disrupted ORF and no *OPN1SW* mRNA expression in the *Mormoops* lineage, but an intact ORF and mRNA expression in the *Pteronotus* lineages, as well as evidence of S-cones in some *Pteronotus* species. Given that the *Mormoops*

and *Pteronotus* lineages diverged ~30 million years ago, and the taxa sampled within the *Pteronotus* lineages diverged ~16 million years ago (*Pavan and Marroig, 2017*), these patterns do not support a disruption in the common ancestor of Mormoopidae as this would have to be followed by different trajectories that led to complete gene loss in one lineage, and partial retained function in the other. The alternative scenario in which each of these cases of loss involved the same mechanism seems highly unlikely given that it would have had to have taken place independently at least four times within the family, with each of the sampled taxa from our study being at a different stage of the gene loss process.

## Differing degree of retention of opsins

In strong contrast to the results from *OPN1SW*, data from proteins and transcripts revealed complete retention of *OPN1LW* across our study species. Such extreme differences in the conservation of color vision genes have previously been reported in other vertebrates (e.g. *Zhao et al., 2009a* and *Zhao et al., 2009b*). Our IHC assays also revealed little evidence of co-localization of both proteins, which is consistent with cone-specific roles. This contrasts with a previous study of two noctilionoid bat species that found that almost all L-cones expressed some S-opsin (*Müller et al., 2009*), although this discrepancy could have arisen from methodological differences. To document fluorescence, the previous study used epifluorescence microscopy, while our study used confocal microscopy. Through its generation of serial optical sections, confocal microscopy typically provides better resolution for co-localization studies. We also found that bat species with S-opsin cones tend to have more L-opsin cones, consistent with both types of cones serving a common functional role.

We also found that S-cone retention varied among conspecifics. In *Pteronotus quadridens*, three of 17 individuals were found to lack S-cones. This heterogeneity could indicate the ongoing degradation of protein synthesis. Indeed, allelic variation has been reported to contribute to opsin variation in diurnal lemurs (*Jacobs et al., 2017*) and has previously been detected in *OPN1SW* in *Pteronotus mesoamericanus* (*Wu et al., 2018*).

## Ecological determinants and agents of selection

Alongside parallel losses of shortwave-sensitive opsins in some noctilionoids lineages, we found strong conservation of S-cones, *OPN1SW* transcription, and protein-coding sequences in around 20 of the species studied. Thus, S-cones appear to still play an important function in these bats. Although the pseudogenization of *OPN1SW*, or loss of transcription had both been previously explained by the use of caves as roosts (*Gutierrez et al., 2018*; *Wu et al., 2018*), our phylogenetic regressions estimated the coefficient for this factor to include 0 (*Figure 5*). Instead, we identified the predominance of fruit consumption as the single most powerful explanatory factor explaining the variation in S-cone presence across the clade, with a similarly positive but not statistically significant effect for *OPN1SW* transcription and protein-coding sequences. Surprisingly, the result for plant-visiting (which includes flower-visiting bats) was not similarly strong and no such result was found for insectivory or cave-roosting. Diet therefore appears to be the primary selective agent for maintaining S-opsin function. We also found that while some predominantly nectarivorous species from both independent nectar-feeding lineages, have lost their S-cones, others have retained them (e.g. *Anoura geoffroyi*). While the loss of S-opsins in flower visiting bats may seem maladaptive, behavioral assays have previously been used to infer that some nectarivorous phyllostomid species appear to be color blind, and thus may be able to utilize UV reflectance to locate flowers via an alternative rod-based mechanism (*Winter et al., 2003*). This suggests that either more than one strategy to locating flowers has evolved among New World leaf-nosed bats, or other non-visual cues are used (e.g. *Gonzalez-Terrazas et al., 2016*).

Since fruit consumption arose as an evolutionary innovation within the Yangochiroptera, selection for this novel niche cannot explain the ancestral or present-day persistence of S-cones in non-frugivorous species. A role in light capture rather than in detecting novel visual cues might explain the density of S- and L-opsin cones in the non-frugivorous lineages sampled, as well as in ancestral bats. The signals of strong purifying selection of all three visual opsin sequences to conserve ancestral function in both frugivorous and non-frugivorous lineages further buttresses interpretation, as it implies there is no detectable relaxation of selection on non-frugivorous lineages, at least among

species for which sequences were available. At the same time, divergent selection in frugivorous and non-frugivorous lineages in *RHO* may further support the importance of light capture, and dim light vision, in relation to novel diets in frugivorous species.

As expected once pseudogenization has occurred, the main difference in molecular selection was between species with an *OPN1SW* ORF and a pseudogene at this locus. Instead of directly reflecting ecological covariates, the process of pseudogenization appears to represent the culmination of a longer term process that starts earlier with cone loss. This highlights post-transcriptional regulation as a more direct response to ecology than pseudogenization of the relevant opsin. Therefore, protein composition should more closely reflect visual ecology than high rates of sequence evolution and pseudogenization in the relevant opsin, as the latter only responds to long-term functional loss. We further tested this inference by modeling the presence of an *OPN1SW* ORF, mRNA, or protein as a function of ecological covariates, finding the strongest ecological association—estimated by higher coefficients—with the presence of S-cones (rather than with earlier steps in protein production).

Recent studies of color vision evolution in New World leaf nosed bats have begun to explore the complex picture of opsin gene loss in the context of selection and ecological factors (*Gutierrez et al., 2018*; *Kries et al., 2018*; *Li et al., 2018*; *Wu et al., 2018*). The detected pseudogenization of *OPN1SW* in infrared sensing vampire bats and in high-duty cycle (HDC) echolocating bats such as *Pteronotus mesoamericanus* (formerly *P. parnellii mesoamericanus*) have led researchers to invoke evolutionary sensory trade-offs as one factor behind the loss of color vision (*Kries et al., 2018*; *Li et al., 2018*; *Wu et al., 2018*). An additional loss was detected in *Lonchophylla mordax* (*Kries et al., 2018*), a nectar bat that roosts in caves, with the roosting preference taken to be driver of color vision loss in this species. In cases in which either no S-opsin losses were inferred in Yangochiroptera, or selection analyses were performed across both Old and New World species simultaneously (*Gutierrez et al., 2018*), it is not possible to interpret the results solely in the context of noctilionoids. Sensory trade-offs, foraging strategy and obligate cave roosting are hypotheses that have previously been applied to loss of S-opsins in Old World bat lineages (e.g. *Zhao et al., 2009a*), however these traits often co-vary within species so the signal may be difficult to disentangle. By allowing us to detect previously 'hidden' opsin phenotypes across noctilionoid species, our approach has allowed us to identify previously undetected ecological factors, that is, fruit consumption, as an explanatory variable of S-opsin retention. Furthermore, the discovery of loss of gene function in non-HDC Mormoopidae, e.g. *Mormoops blainvillei* and *Pteronotus davyi*, also call into question the sensory trade-off hypothesis within this family.

## Study limitations and alternative interpretations

The surprising diversity in S-opsin retention recorded in this study was seen across divergent species, congeners, and even conspecifics. Although such patterns could also arise from methodological issues, some of our findings appear consistent with emerging trends. For example, within *Pteronotus* and the Mormoopidae family as a whole, there is increasing evidence to support an extremely complex evolutionary history of S-opsins (*Gutierrez et al., 2018*; *Simões et al., 2018*; *Wu et al., 2018*). In comparison, methodological artifacts are less easy to rule out as causes of variation among *Carollia* spp. given that S-cone presence in two species was inferred from either recently collected field specimens, or a mixture of both field and museum specimens, while S-cone absence in a third species was based on museum specimens collected in 1968 and 1972.

Despite this, the utility of long-term fixed specimens for immunohistochemical staining of proteins (e.g. vimentin and GFAP) has been described previously (*Hühns et al., 2015*; *Thewissen et al., 2006*), including visual opsins in some museum specimens (~20 years old and using the same S-opsin antibody as used in the current study) (*Nießner et al., 2016*). In line with this, we were able to recover good IHC staining for both S- and L-cones from our oldest sampled museum specimen with a confident date, an *Artibeus fraterculus* collected in 1921 (see Figure supplement S3). We note, however, that our S-opsin assays were inconclusive for six species (*T. brasiliensis*, *P. hastatus*, *S. tildae*, *S. ludovici*, *P. dorsalis* and *C. villosum*) represented by museum samples due to low signal-to-background ratios, but L-opsin assays were successful in these species. The ability to detect protein in these, and other, museum specimens is a function of the condition of the retina and the density of the cones in question. Because L-cones are present at higher densities than S-cones (this study and *Müller et al., 2009*), we were able to more readily detect L-cones even in more poorly preserved

retina. Thus, while museum collections are rarely used in protein studies, relative to their use, for example, in genetic and genomic studies (e.g. *Bailey et al., 2016*; *Nießner et al., 2016*), they offer great potential for a range of comparative studies provided that caution is exercised (e.g. *Hedrick et al., 2018*). These benefits apply particularly to groups that cannot be sampled in the wild for ethical, conservation and logistic reasons (e.g. *Russo et al., 2017*). In our study, restrictions on sampling necessitated our comparisons of conspecifics collected from different countries for genetic and protein assays. For this reason, we cannot rule out geography as a source of variation, and it is noteworthy that one focal taxon, *Pteronotus parnellii* was recently recognized as a species complex with a strong phylogeographic divergence (*Pavan and Marroig, 2017*). Although other focal bat species have not been split in this way, many have wide ranges and their genetic diversity may be underestimated (*Clare et al., 2011*). In another instance, our fresh specimen of *M. molossus* from Belize and two older museum specimens from 1968 from Uruguay all lacked S-cones, whereas a published record of an individual from an unknown geographical locality showed S-cone presence (*Nießner et al., 2016*). These patterns of S-cone presence, and indeed those across the entire noctilionoid tree, suggest that losses may arise across populations of the same species.

We must also consider whether our results might arise from methodological artifacts related to the short read data. For example, low gene expression can limit the number of representative reads in RNA-Seq datasets for transcript assembly (e.g. *Zhao et al., 2011*) and this caveat likely applies to shortwave-sensitive opsins that show low S-cone densities, inferred loss of function in some taxa, and also low expression levels based preliminary expression analyses (data not shown). To cross-validate our assembled transcripts, whenever possible we compared our sequences to published data from PCR amplicons, RNA-Seq, and/or genome datasets. For example, for *Monophyllus redmani*, *Trachops cirrhosus* and *Pteronotus parnellii* we were able to confirm that our assembled transcripts matched published assembled mRNA contigs, PCR amplicons and genomic sequences, respectively (*Gutierrez et al., 2018*; *Tsagkogeorga et al., 2013*; *Wu et al., 2018*). For *Mormoops blainvillei* and *Macrotus waterhousii*, the absence of *OPN1SW* mRNA transcripts and S-cones was supported by disrupted ORFs in published genome datasets, as well as the highly divergent and fragmented transcripts recovered by a recent study, which our visual inspections suggest may be due to cross contamination or misassembly (*Gutierrez et al., 2018*). Therefore, several lines of evidence support loss of function of OPN1SW in these taxa. For *C. micropus*, our inferred intact ORF is based on PCR of ~100 codons (also seen in *B. pumila*) and was supported by sequence data from two closely related species from the Natalidae (*Emerling et al., 2015*; *Simões et al., 2018*).

For several species, our transcriptomic analysis detected multiple *OPN1SW* mRNA transcripts variants, characterized by retained introns and missing exons. We are able to confirm the species-specific intronic sequences of several of the species due to recently available genomes and gDNA PCRs (*Kries et al., 2018*; *Li et al., 2018*; *Tsagkogeorga et al., 2013*; *Wu et al., 2018*). The observed retention of introns in *OPN1SW* mRNA as well as the expression of pseudogenized opsin mRNA are both supported by earlier studies (*David-Gray et al., 2002*; *Schweikert et al., 2016*); however, alternative scenarios for these findings could include gDNA contamination, sequencing of immature mRNA or low-level cross contamination resulting in the assembly of highly divergent transcripts. Finally, for most species our mRNA evidence is based on one individual. However, the lack of a clear relationship between RIN score or sequencing depth and presence of OPN1SW mRNA (data not shown), together with the presence of the two other visual opsins in all samples, suggests this should be sufficient.

## Other studies of parallel loss and associated mechanisms

Our findings provide important insights into how parallel losses occur in response to diverse ecological demands, as well as how several alternative molecular routes may lead to the same phenotype. There are other examples of parallel loss from pelvic reduction in sticklebacks (via repeated changes in a *Pitx1* enhancer), color and vision in *Astyanax* cavefish (via loss of function of *Oca2*), trichomes in *Drosophila* spp., and floral pigments in Iochrominae (*Chan et al., 2010*; *Larter et al., 2018*; *McGregor et al., 2007*; *Protas et al., 2006*), but few of these have examined as many species across as many steps of phenotype production. Our data and those from other recent studies on bat opsins associate independent losses of S-cones with diverse adaptations (e.g. shifts in diet, roosting ecology and sensory traits), and are therefore consistent with multiple, distinct ecological demands leading to the same phenotype. Hence, our findings are also consistent with the hypothesis that UV

vision represents a genetic 'hot spot' of evolution (*Hoekstra and Coyne, 2007*; *Martin and Orgogozo, 2013*; *Stern and Orgogozo, 2008*), along an evolutionary line of least resistance (*Schluter, 1996*). Therefore, by documenting a range of molecular routes to functional degradation, this study supports the hypothesis that vision is a highly evolvable trait that repeatedly and rapidly changes in response to diverse selective demands.

In conclusion, our findings reveal that assessments of visual perception based purely on genotypic analyses of either opsin sequences or RNA transcripts can be misleading, and may even obscure the evolutionary processes and ecological agents of selection. Although variation in the complement of photoreceptors across vertebrates is usually explained by disruptions to the protein-coding sequence (e.g. *Mundy et al., 2016* and *Zhao et al., 2009a*), findings of mismatches between genotype and phenotype also indicate a role for transcriptional and even translational control in this process. It follows that because routes of gene loss are mainly studied at the genetic level or, in fewer cases, at the transcriptomic level, the input of changes in translation and other connections between the genetic, transcriptomic, and proteomic levels may be being underestimated. More broadly, our results highlight the importance of rapid trait loss in evolution, with apparent shifts in translation and transcription that precede pseudogenizing changes in ORFs. As genotype-centered analyses would miss important functional changes, our study also illustrates the importance of probing multiple levels of protein synthesis.

# Materials and methods

**Key resources table**

| Reagent type (species) or resource | Designation | Source or reference | Identifiers | Additional information |
|---|---|---|---|---|
| Antibody | sc-14363, goat anti-OPN1SW | Santa-Cruz Biotechnologies | RRID: AB_2158332 | (1:1000) |
| Antibody | ab5405, rabbit anti-opsin red/green | Millipore Ibérica, Madrid | RRID: AB_177456 | (1:750) |
| Antibody | donkey anti-goat Alexa Fluor 568 | Thermofisher | RRID:AB_2534104 | (1:500) |
| Antibody | donkey anti-rabbit Alexa Fluor 647 | Thermofisher | RRID: AB_10891079 | (1:500) |
| Software, algorithm | FIJI | https://fiji.sc/ | | |
| Commercial assay or kit | RNeasy Mini kits | Qiagen | | |
| Commercial assay or kit | Qiagen DNeasy Blood and Tissue Kit | Qiagen | | |

## Species sampling and tissue preparation

We obtained eye tissue from 59 New World bat species, of which 49 were collected from the wild and 34 from the American Museum of Natural History (AMNH), with 24 species common to both sources (*Supplementary file 1*). Our sampling was designed to maximize taxonomic coverage and include as many replicates as possible within ethical and regulatory limits. Unlike lab animals such as mice or rats, most bat species have just one offspring per year (*Wilkinson and South, 2002*), limiting the rate of recovery from adult mortality. All wild bats were captured with traps set in forests and/or at cave entrances, were handled, and then euthanized by isoflurane overdose, under appropriate research and ethical permits (see Appendix 1).

## RNA sequence analysis

Intact eyes were placed in RNAlater and incubated at 4°C overnight and then frozen. Total RNA was isolated using Qiagen RNeasy Mini kits with the addition of DTT and homogenization using a Qiagen TissueLyser. Following QC, total RNA from each individual was used to construct a cDNA library using the Illumina TruSeq RNA v2 kit. Pooled libraries were sequenced (NextSeq 500). Eye transcriptomes were generated for 46 individuals (39 species) including biological replicates of *Pteronotus parnellii* ($n = 4$), *Artibeus jamaicensis* ($n = 4$) and *Phyllops falcatus* ($n = 2$). Raw reads were trimmed, and clean reads were assembled with Trinity v.2.2.0 (*Grabherr et al., 2011*) (see Appendix 1).

We tested for the presence of the three focal gene transcripts (*RHO*, *OPN1SW* and *OPN1LW*) in each bat transcriptome using a reciprocal best hit blast approach against the full set ($n = 22,285$) of human protein-coding genes from Ensembl 86 (*Yates et al., 2016*). To confirm the absence of

*OPN1SW* sequence, we performed additional steps in several species. First, we cut, *trans*-chimeras, which can prevent detection by reciprocal blast (*Yang and Smith, 2013*), and repeated the reciprocal blast. Second, we manually screened sequences that were initially identified as matching *OPN1SW*, but did not pass initial blast filtering (see Supplementary Information). Recovered opsin gene sequences have been submitted to GenBank (accession numbers MK209460 - MK209505 [RHO]; MK209506 - MK209551 [OPN1LW]; and MK209552 - MK209592 [OPN1SW]).

Additionally, for each individual RNA dataset we manually aligned all assembled transcripts, that passed the tblastn step of the reciprocal blast for *OPN1SW*, together with individual exons and introns obtained from the *Myotis lucifugus* structural annotation downloaded from Ensembl. Finally, we obtained all *OPN1SW* DNA and mRNA sequences currently available for our study species from GenBank, produced by recently published studies or genomes (*Gutierrez et al., 2018*; *Kries et al., 2018*; *Li et al., 2018*; *Wu et al., 2018*; *Zepeda Mendoza et al., 2018*). This data was used to confirm either our mRNA assemblies or the intronic sequences, and also to infer ORF status for species in which we had protein data for but were not able to obtain tissue for RNA-seq (e.g. *Eptesicus fuscus*, *Pteronotus davyi*, *Diaemus youngi*, *Phyllostomus discolor*, *Sturnira lilium*)

## Immunohistochemistry (IHC) and photoreceptor quantification

### IHC assays

Specimens for IHC were obtained from the wild (fresh) and from collections of the AMNH (preserved). Given the variability of the age of the preserved museum specimens, the initial fixation method is not always known. However, it is most likely they would have been initially fixed with formaldehyde/formalin and then stored in 70% ethanol. Fresh eyes were fixed overnight at 4°C in 4% paraformaldehyde (PFA) in phosphate-buffered saline (PBS), transferred into 1X PBS, and then stored in 1% PFA in 1X PBS at 4°C until further processing. Preserved eyes had been collected previously and stored in 70% ethanol at room temperature for varying lengths of time (see *Supplementary file 1*). Preserved eyes were rehydrated through an ethanol series (70% 20 min, 50% 20 min, 20% 20 min) in 1X PBS and then stored in 1% PFA at 4°C in 1X PBS. Prior to processing, retinas were dissected from eyeballs and flattened by making three or four radial incisions from the outside of the retina inwards, with the deepest cut in the nasal pole. Immunodetection was carried out following standard procedures described in *Ortín-Martínez et al., 2014*. Briefly, retinas were permeated with two washes in PBS 0.5% Triton X-100 (Tx) and frozen for 15 min at −70°C in 100% methanol. Retinas were then thawed at room temperature, rinsed twice in PBS-0.5%Tx and incubated overnight at 4°C in the appropriate mixture of primary antibodies diluted in blocking buffer (PBS, 2% normal donkey serum, 2%Tx). The next day, retinas were washed four times in PBS-0.5Tx and incubated for 2 hr at room temperature in secondary antibodies diluted in PBS-2%Tx. Finally, retinas were thoroughly washed four times in PBS-0.5%Tx and, after a last rinse in PBS, mounted scleral side up on slides in anti-fading solution (Prolong Gold Antifade, Thermofisher). For each species, IHC was performed on at least three retinas from two individuals (for details see *Supplementary file 1*).

### IHC - antibodies and working dilutions

The following primary antibodies were used: goat anti-OPN1SW, 1:1000 (RRID: AB_2158332, sc-14363, Santa-Cruz Biotechnologies, Heidelberg, Germany; detects S-opsin protein) and rabbit anti-opsin red/green, 1:750 (RRID: AB_177456, ab5405, Millipore Ibérica, Madrid, Spain; detects L-opsin protein). Sc-14363 is an affinity-purified goat polyclonal antibody raised against a 20-amino-acid synthetic peptide mapping within amino acids 1 to 50 of human blue-sensitive opsin, and AB5405 was raised in rabbit against the last 42 amino acids of the C-terminus of recombinant human red/green opsin (*Gaillard et al., 2009*). These antibodies have been used successfully in many groups, including rodents, artiodactyls, bats, and birds (e.g. *Gaillard et al., 2009*, *Müller et al., 2007* and *Nießner et al., 2016*). The following secondary antibodies were used at a 1:500 dilution: donkey anti-goat Alexa Fluor 568 (RRID: AB_2534104) and donkey anti-rabbit Alexa Fluor 647 (RRID: AB_10891079) (Thermofisher). In addition, we created amino acid alignments of the peptide regions thought to correspond to the antibody epitopes across the bat species studied to assess sequence variation.

## Quality control of retinal IHC

We used a strong quality control protocol to ensure that we could interpret an absence of labelling as a true loss of S-opsin protein. Given the variable age and preservation of museum specimens, we evaluated the anatomical preservation of the retina during dissection and excluded specimens (data not shown) if the retina was: (1) attached to the crystalline lens, poorly preserved, impossible to dissect/damaged, (2) highly fragile, poorly preserved, disintegrated/damaged upon dissection, and (3) intact or preserved in large pieces but exhibited shrinkage and/or an orange color characteristic of tissue degradation. When necessary, we also slightly modified the IHC protocol for some museum samples. Specifically, since museum samples where already permeabilized by their storage in ethanol, we reduced the number of PBS-0.5%Tx washes and removed the methanol permeabilization step at −70C. With these quality-control measures in place, and given the consistency of the detection of our chosen antibodies across all bats and other mammals (*Müller et al., 2009*; *Müller et al., 2007*; *Ortín-Martínez et al., 2014*), and the number of replicates and individuals we examined, we are confident in our interpretation that no labeling indicate a true loss of the respective cone type. In addition, we created amino acid alignments of the peptide regions thought to correspond to the antibody epitopes across the bat species studied to gain a measure of the sequence variation at these points.

## IHC - photoreceptor quantification

Flat-mounted retinas were photographed using a 20X objective on a confocal microscope (LSM710; Zeiss Microscopy). 564 and 633 lasers were used to excited Alexa 568 and Alexa 647 dyes, labelling S- and L- opsins, respectively. Each entire retina was completely imaged using $512 \times 512$ pixel tiles. For each retina, each tile was then Z-stacked and automatically counted using a 3D object counter plugin using Fiji (ImageJ). The accuracy of this automatic approach was verified by manually counting three biological replicates of five bat species, by two different people. For each retina quantified, the density was calculated for each tile and then averaged for each individual (total count was average over three individuals) and for each species (by averaging the average of the three individuals). The spatial distribution of L- and S-cone density was visualized for the following 14 species: *Artibeus jamaicensis*, *Artibeus phaeotis*, *Carollia sowelli*, *Sturnira lilium*, *Monophyllus redmani*, *Erophylla sezekorni*, *Glossophaga soricina*, *Brachyphylla nana pumila*, *Desmodus rotundus*, *Pteronotus quadridens*, *Mormoops blainvillei*, *Macrotus waterhousii*, *Gardnerycteris crenulatum* and *Phyllops falcatus* (see *Figure 2*, Table S2 in *Supplementary file 2*).

## Opsin gene evolution

We used aligned sequences from the transcriptomes of 38 species together with those from six noctilionoid genomes (*Zepeda Mendoza et al., 2018*) to estimate rates of molecular evolution of visual opsin genes (*OPN1SW*, *OPN1LW*, and *RHO*) in focal bats. First, we tested for divergent selection modes among species that had S-opsin cones, lacked the S-opsin cones but had an intact mRNA sequence, and those that lacked the S-opsin cones but either did not have *OPN1SW* transcripts or had a pseudogenized *OPN1SW* sequence (*Figure 5—figure supplement 1*) using the Branch Model 2 of codeml in PAML 4.8a (*Yang, 2007*). Second, we applied the same approach to test divergent selection modes between frugivorous and non-frugivorous bat species (*Figure 5—figure supplement 1*; gene alignments have been submitted to DRYAD http://dx.doi.org/10.5061/dryad.456569k).

## Ecological correlates of cone presence and density

To determine whether cone phenotypes are explained by dietary specialization, we applied the hierarchical Bayesian approach implemented in the R packages MCMCglmm and mulTree (*Guillerme and Healy, 2014*; *Hadfield, 2010*), using a sample from the posterior distribution of phylogenies of New World noctilionoids grafted onto the phylogeny of bats (*Rojas et al., 2016*; *Shi and Rabosky, 2015*). We modeled S-cone presence with species as observations as function of diet represented by four variables ($n_{species}$ = $n_{observations}$ = 50). Since all predictor variables correspond to the presence or absence of a given diet or roosting habit, the coefficients of the resulting models were used to compare the strength of the association between the *OPN1SW* genotype or phenotype and the ecological covariate. Since this modeling approach neither tests against a null hypothesis of no

effect, nor assumes the point estimates –in this case the mans by ecological group– are stationary, there is no requirement to adjust for multiple comparisons (*Gelman et al., 2012*). Using the predictor variable from the best model for presence/absence of S-opsin cones, or the presence of S-cones as a factor, we then repeated this approach to explain L-cone density across individuals within species ($n_{species}$ = 14, $n_{observations}$ = 33, see Supplementary Information). To normalize the response data (density), we transformed by taking the natural logarithm of the cone density estimate. These analyses took advantage of the hierarchical structure of observations of density replicates clustered within species, with estimates of variance between species (corresponding to the phylogenetic regression), and residual variance remaining between observations. Given this data design, the estimate of the mean density per-group (i.e. frugivory/non, or presence/absence of S-cones) accounts for both between and within species variance. The R code for all regression models is available from DRYAD http://dx.doi.org/10.5061/dryad.456569k

## Acknowledgements

We thank M Agudo-Barriuso, PK Ahnelt, L Peichl, G Tsagkogeorga and staff at the Queen Mary Genome Centre for advice, lab assistance and protocols. For help with permits and field support in the Dominican Republic, we thank J Almonthe, ME Lauterbur, YM León, MS Nuñez, and J Salazar; in Peru, F Cornejo, J Pacheco, J Potter, H Portocarrero, MK Ramos, E Rengifo, JN Ruiz, C Tello; and in Puerto Rico, A Rodriguez-Duran, and N Ann. For access to museum specimens, we thank N Simmons (American Museum of Natural History). For help with permits and field support in Belize, we thank M Howells and the Lamanai outpost lodge staff, N Simmons and B Fenton. For help with permits and field support in Costa Rica, we thank B Matarrita, M Porras, LB Miller, A Kaliszewska, S Santana and the La Selva Biological Station staff. For providing bat images we thank E Clare, and D Rojas for providing roosting ecology data. Results in this paper were obtained using the high-performance LI-RED computing system at the Institute for Advanced Computational Science at Stony Brook University, the Indiana University Mason server funded by NSF-DBI 1458641, and Queen Mary's MidPlus computational facilities supported by QMUL Research-IT and funded by EPSRC grant EP/K000128/1. This research was conducted under research permits VAPB-01436 in the Dominican Republic, 0002287 in Peru, and R-018-2013-OT-CONAGEBIO in Costa Rica.

## Additional information

### Funding

| Funder | Grant reference number | Author |
| --- | --- | --- |
| National Science Foundation | 1442314 | Karen E Sears<br>Alexa Sadier<br>Kun Yun |
| National Science Foundation | 1442142 | Kalina TJ Davies<br>Laurel R Yohe<br>Paul Donat<br>Liliana M Dávalos<br>Stephen J Rossiter |
| European Research Council | 310482 | Stephen J Rossiter<br>Kalina TJ Davies |
| National Science Foundation | 1442278 | Elizabeth R Dumont<br>Brandon P Hedrick |

The funders had no role in study design, data collection and interpretation, or the decision to submit the work for publication.

### Author contributions

Alexa Sadier, Conceptualization, Data curation, Formal analysis, Investigation, Visualization, Methodology, Writing—original draft, Writing—review and editing; Kalina TJ Davies, Laurel R Yohe, Conceptualization, Formal analysis, Validation, Investigation, Visualization, Methodology, Writing—original draft, Writing—review and editing; Kun Yun, Investigation, Visualization; Paul Donat,

Investigation, Methodology; Brandon P Hedrick, Formal analysis, Investigation, Methodology, Writing—original draft, Writing—review and editing; Elizabeth R Dumont, Conceptualization, Formal analysis, Supervision, Funding acquisition, Validation, Investigation, Visualization, Methodology, Writing—original draft, Project administration, Writing—review and editing; Liliana M Dávalos, Stephen J Rossiter, Karen E Sears, Conceptualization, Data curation, Formal analysis, Supervision, Funding acquisition, Validation, Investigation, Visualization, Methodology, Writing—original draft, Project administration, Writing—review and editing

### Author ORCIDs

Alexa Sadier (iD) http://orcid.org/0000-0002-9907-3714

Kalina TJ Davies (iD) http://orcid.org/0000-0002-4258-4775

Laurel R Yohe (iD) http://orcid.org/0000-0003-1567-8749

Brandon P Hedrick (iD) http://orcid.org/0000-0003-4446-3405

Elizabeth R Dumont (iD) http://orcid.org/0000-0002-7809-388X

Liliana M Dávalos (iD) http://orcid.org/0000-0002-4327-7697

Stephen J Rossiter (iD) http://orcid.org/0000-0002-3881-4515

Karen E Sears (iD) http://orcid.org/0000-0001-9744-9602

### Ethics

Animal experimentation: This study was performed in strict accordance with the recommendations in the Guide for the Care and Use of Laboratory Animals of the National Institutes of Health. All of the animals were handled according to approved institutional animal care and use committee (IACUC) protocols 14199 at UIUC, 2017-093 at UCLA, and 2013-2034-NF-4.15.16-BAT and 2014-2090-R1-1.20.17-Bat at SBU. Every effort was made to minimize suffering.

### Decision letter and Author response

Decision letter https://doi.org/10.7554/eLife.37412.022

Author response https://doi.org/10.7554/eLife.37412.023

## Additional files

### Supplementary files

• Supplementary file 1. Specimen and sampling information for the tissues used by this study. Unknown collection month indicated by 'unk', and (?) indicates uncertainty in the museum specimen collection date.

DOI: https://doi.org/10.7554/eLife.37412.015

• Supplementary file 2. Results of molecular evolution branch analyses for each of the three opsin genes tested for differences in rates of nonsynonymous to synonymous substitutions ($\omega$) for lineages that lack the S-cone and lineages that have retained the S-cone. Grey boxes indicate the preferred model inferred from a likelihood ratio test. *lnL*: log-likelihood; *np*: number of parameters; *TL*: tree length; *k*: kappa (transition/transversion ratios); LR: likelihood ratio; *p*: *p*-value of likelihood ratio of alternative relative to null for each test

DOI: https://doi.org/10.7554/eLife.37412.016

• Transparent reporting form

DOI: https://doi.org/10.7554/eLife.37412.017

### Data availability

Sequencing data have been deposited in GenBank in the Nucleotide Database. The accession numbers are as follows: RHO: MK209460 - MK209505; OPN1LW: MK209506 - MK209551; OPN1SW: MK209552 - MK209592. The GenBank numbers for the OPN1SW PCR sequences are MK248618 - MK248630. Gene alignments and R code for regressions are available via Dryad (http://dx.doi.org/10.5061/dryad.456569k).

The following dataset was generated:

| Author(s) | Year | Dataset title | Dataset URL | Database and Identifier |
|---|---|---|---|---|
| Sadier A, Davies KTJ, Yohe LR, Yun K, Donat P, Hedrick BP, Dumont ER, Davalos LM, Rossiter SJ, Sears KE | 2019 | Gene alignment data from Multifactorial processes underlie parallel opsin loss in neotropical bats | http://dx.doi.org/10.5061/dryad.456569k | Dryad Digital Repository, 10.5061/dryad.456569k |

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

# Appendix 1

DOI: https://doi.org/10.7554/eLife.37412.018

## Species sampling

We obtained eye tissue samples from a total of 263 eyes from 232 individuals, representing 59 bat species from seven families. This includes three families from the focal Noctilionoidea superfamily and four closely related outgroup families. Specimens used for this study were either wild-caught animals or obtained from museum collections (see *Supplementary file 1* for full species and permit information). Wild-caught specimens from 49 bat species were collected following the approved IACUC protocols and site-specific permits. Bats were sampled from wild populations, and caught using mist nests set in forests and/or at cave entrances. Animals were handled following IACUC and site-specific protocols to minimize stress and were euthanized using an excess of isoflurane. Fresh bat specimens used for RNA-Seq analyses ($n_{RNA\text{-}Seq}$ = 45) were sampled in the Dominican Republic, Peru and Costa Rica, and those for immunohistochemistry were sampled in the Dominican Republic, Puerto Rico, Belize, and Trinidad. Additional eye samples from 34 species were dissected from specimens from the American Museum of Natural History (AMNH). For immunohistochemistry, five bat species had replicates that were both wild-caught and from museum collections and exhibited the same phenotype, highlighting the robustness of the experiments. Due to preservation methods, the AMNH samples were only suitable for immunohistochemistry (IHC) and so were not included in the transcriptomic study (see *Supplementary file 1* for AMNH specimen identification numbers).

## RNA sequence analysis

Shortly after death, intact eyes were excised and placed in RNAlater and incubated at 4°C overnight before being stored at −180°C in vapor-phase liquid nitrogen, or at −80°C in a freezer. Frozen eye tissues were placed in Buffer RLT with added Dithiothreitol (DTT) then homogenized using a Qiagen TissueLyser. Total RNA was then extracted using Qiagen RNeasy Mini kits following the manufacturer's protocol. Following extraction, RNA integrity was assessed using an Agilent 2100 Bioanalyzer and RNA concentration was measured using a Qubit Fluorometer. Library preparation was performed using Illumina TruSeq RNA Sample Preparation v2, with 500 ng of total RNA used for each sample. Constructed libraries were pooled and sequenced on the NextSeq500 High Output Run (150 cycles) to give 2 × 75 base-pair (bp) paired-end (PE) reads at the Genome Centre, Queen Mary University of London. Using the above approach, we sequenced eye transcriptomes from 45 individuals, representing 38 bat species; including biological replicates for three species (*Pteronotus parnellii: n* = 4; *Artibeus jamaicensis: n* = 4 and *Phyllops falcatus: n* = 2).

We assessed the quality of the short-read data with FastQC v.0.11.5 (http://www.bioinformatics.bbsrc.ac.uk/projects/fastqc). Raw reads were trimmed with Trimmomatic-0.35 (*Bolger et al., 2014*), with the following settings LEADING:3 TRAILING:3 SLIDINGWINDOW:4:15 MINLEN:36. Cleaned reads were assembled into de novo transcriptomes with Trinity v.2.2.0 (*Grabherr et al., 2011*), using default parameters.

## Opsin gene annotation and examination of CDS

We used a combination of transcriptomic and genomic data to establish if the coding sequences (CDSs) of the three focal genes (*OPN1SW*, *OPN1LW* and *RHO*) were intact, and whether or not the mRNA of these genes was expressed in the eyes of the bats under study. We used a reciprocal best hit blast approach to establish whether or not transcripts corresponding to the three visual pigments (*OPN1SW*, *OPN1LW* and *RHO*) were present in each of the assembled transcriptomes. Sequences representing the protein products encoded by 22,285 human protein coding genes were downloaded from Ensembl 86 (*Yates et al., 2016*), for each gene product only the longest protein sequence was retained. These sequences were then used as tblastn (blast +v0.2.2.29) queries against each of the 45 bat

transcriptome databases, the top hit was kept with an e-value cut-off <1e$^{-6}$. Reciprocal blasts were then carried out using blastx (blast +v0.2.2.29), with bat transcripts as queries against the human protein database, again only keeping the top hit and e-value <1e$^{-6}$. Percentage coverage of each bat hit against the human protein was calculated with the perl script analyze_blastPlus_topHit_coverage.pl available in Trinity utils. Candidate coding sequences (CDSs) were then extracted from the transcriptome assembly based on the blast coordinates using a custom perl script.

For transcriptome assemblies in which the *OPN1SW* sequence was not initially recovered, we undertook a number of additional steps to confirm that the sequence was not present. Firstly, as de novo transcriptome assemblies can create erroneous chimeric transcripts that may affect reciprocal blast results we followed the approach of (*Yang and Smith, 2013*) to reduce the number of *trans*-self-chimeras. Briefly, this involves performing a blastx search of the bat queries against the human protein database with an e-value cutoff of 0.01. Hits that met the default parameters of identity ≥30% and length ≥100 base-pairs are used to then search for either self-chimeras or multi-gene chimeras. Detected putative chimeras are then cut into segments, and retained if >100 base-pairs. This approach is only able to screen for *trans*-chimeras. Following chimera detection, the initial reciprocal blast described above was repeated. Lastly, we manually re-blasted all sequences that were initially identified as matching *OPN1SW*, but did then not pass the stringent reciprocal blast procedure.

## Verification of OPN1SW sequence with PCR

For three species we amplified the genomic region spanning exons 3–4 of *OPN1SW*, from DNA extracted from the same individuals used to generate the RNA-Seq data. DNA extractions were carried out using Qiagen DNeasy Blood and Tissue Kit (69504). We used previously published primers (*Zhao et al., 2009a*). PCR mixtures consisted of 12.5 µl EconoTAQ Master Mix 2x, 3 µl of each primer (10 µM), 4 µl of genomic DNA (>100 ng). On an Eppendorf Mastercycler ProS, PCR was carried out with a single cycle at 94°C for 2 min followed by 35 cycles of 94°C for 30 s, annealing temperature of 55°C for 40 s, 72°C for 1 min 30 s, and finally a single cycle of 72°C for 10 min. PCR products were visualized on a 1% TBE-agarose gel. PCR product was cleaned up using Agencourt AMPure XP and submitted for cycle sequencing. Sequences have been submitted to GenBank MK248618 - MK248630.

## Opsin gene evolution

We obtained protein-coding sequences for *OPN1LW* and *RHO* from the transcriptomes of all 39 species and included sequence data for *Eptesicus fuscus* (from GenBank) and *Phyllostomus discolor* (mPhyDis1_v1.p.fasta, available from GenomeArk vgp.github.io), for a total of 41 species. For *OPN1SW*, we obtained sequence data for 31 species from the transcriptome data sets. In a number of cases coding sequences from multiple contigs were manually joined together to produce a full-length CDS. We then supplemented the *OPN1SW* nucleotide sequences with those extracted from genome data using a combination of blastn and bl2seq on five noctilionoid bat genomes (*Artibeus jamaicensis, Desmodus rotundus* (*Zepeda Mendoza et al., 2018*), *Lionycteris spurrelli, Macrotus waterhousii, Mormoops blainvillei, Phyllostomus discolor* and *Noctilio leporinus*) using the *Miniopterus natalensis OPN1SW* (XM_016213323.1) sequence as a query. The *L. spurrelli* and *M. waterhousii* genomes were sequenced by the Rossiter Lab, and the *A. jamaicensis, M. blainvillei* and *N. leporinus* genomes were made available by the Broad Institute, and *P. discolor* as above. We also obtained the *OPN1SW* sequences from *E. fuscus* from GenBank. The extracted and aligned sequences are available from DRYAD (http://dx.doi.org/10.5061/dryad.456569k).

Sequences for *OPN1LW* and *RHO* were aligned using MUSCLE v3.8.425 (*Edgar, 2004*) as translated amino acids to keep the sequences in frame. The software implementation of our model requires the alignment to be in frame without any stop codons, therefore, stop codons at the end of the reading frame were removed from the alignment. Due to stop codons and indels, nucleotide sequences for *OPN1SW* were aligned by eye, with columns containing disruptions to the reading frame being removed to keep the remaining sequences in frame. Hypervariable regions at the beginning or end of sequences that may be caused assembly

errors were masked by 'Ns' in two cases (*Phyllops falcatus* and *Phyllonycteris poeyi*). Additionally, premature stop codons were masked with 'Ns' and columns containing insertions that shifted the translation frame were deleted to keep codons in frame.

We tested for whether there were differences in rates of molecular evolution in the three opsin genes by estimating the ratio of the rates of nonsynonymous to synonymous substitutions ($\omega$) for different branch classes. We set up two frameworks: S-cone presence and diet. When looking at S-cone presence variation, in our 2-branch class test, we estimated differences for branches that lacked the S-cone protein ($\omega_{S-cone.absent}$), and those that had the S-cone protein present ($\omega_{background}$). We also designed a three branch-class test in which we estimated different rates for bats with S-cones ($\omega_{background}$), bats that lack S-cones but have an intact reading frame for the *OPN1SW* transcripts ($\omega_{OPN1SW.intact}$), and bats that lack S-cones and *OPN1SW* is a pseudogene ($\omega_{OPN1SW.pseudo}$). If bats that lack the S-cone experience relaxed selection, we expect higher rates of $\omega$ in bats without the S-cone in the *OPN1SW* gene, but no differences among groups in the other two opsins. The sequences from lineages for which no S-cone data was available were removed from this analysis ($n = 8$). Finally, we tested if there were differences in rates in frugivorous lineages ($\omega_{frugivore}$) and all other bats ($\omega_{background}$). **Figure 5—figure supplement 1** depicts branches labeled with respective branch classes. Frugivory data was available for all lineages, and thus all available sequences were used in this analysis.

These analyses were performed using the branch model implemented in the codeml routine of PAML 4.8a (**Yang, 2007**). Differences among branches were compared against estimates for a single $\omega$ for all branches ($\omega_{background}$). We used a likelihood ratio test to compare the best-fit model for each opsin gene. The analysis used the species topology that merged a recently published phylogeny of all bats (**Shi and Rabosky, 2015**) with that of a recently published noctilionoid tree (**Rojas et al., 2016**). The tree was trimmed using the geiger v. 2.0.6 package in R (**Harmon et al., 2008**).

## Ecological correlates of cone presence and density

A hierarchical Bayesian approach was used to relate the ecological factors to the presence of an *OPN1SW* ORF, mRNA, or S-cone while accounting for the phylogenetic correlation between observations from different species. A hierarchical approach is often called a mixed model in the literature, with cluster-specific effects called 'random', and sample-wide effects called 'fixed'. As different fields apply 'random' and 'fixed' to different levels of the hierarchy, here we adopt the language of cluster-specific and sample-wide effects (**Gelman, 2005**). The effect of species was quantified by including species as a cluster-specific, or random effect in the R package MCMCglmm (**Hadfield, 2010**). Additionally, to address variation among different estimates of phylogeny, we used the R package mulTree (Guillerme & Healy, 2014) to run the Bayesian models across a sample of trees obtained from the posterior of phylogenies.

The first set of models explained presence or absence of an *OPN1SW* ORF, mRNA, or S-cone as a function of one in a series of ecological dummy variables coded as prevalence or non-prevalence of plant materials, fruit, or other (insects or small vertebrates) items in diet, or whether known roosts included caves or not. In the sample-wide or fixed portion of these models, observations $y$ for each species from one to $i$ for each ecological category one to $j$ correspond to a single-trial binomial response of the probability of observing the *OPN1SW* genotype or phenotype given by $pr_i$ such that:

$$y_i \sim dbern(pr_i)$$

$$logit(pr_i) = a + b.ecology[ecology_j]$$

Models for each of the dietary categorizations were then compared using the estimated coefficients. For dummy predicators indicating presence/absence, the absolute coefficient on the presence indicates the strength of the association with the response (in this case the genotype or phenotype). The predictor variable identified as the most strongly associated with the presence/absence of S-cones was then used in subsequent analyses of cone density.

Analyses of the sample-wide or fixed portion of cone density modeled the natural logarithm of the cone density $y$ for each observation $i$ as a function of dummy predictor variables defined by an diet group $j$, or S-cone group $k$. $ln(y_i)$ was modeled as a random, normally distributed variable with mean mu and variance, as below:

$$ln(y_i) \sim dnorm(mu, variance)$$

$$ln(y_i) = a + b.diet[diet_j]$$

$$ln(y_i) = a + b.S.cone[S.cone_k]$$

Unlike the presence/absence analyses, these response variables were normally distributed, with the sample-wide portion of the model accounting for the effect of diet or S-cone presence on L-cone density. The cluster-specific or random effect accounted for both the relationships between species and the clustering of observations when more than one measurement was taken for each species.

To estimate the covariation arising from phylogeny for all species analyzed, we used the phylogeny of *Shi and Rabosky (2015)* for non-noctilionoids and as a base tree. For New World noctilionoids, a posterior sample of 100 trees from the phylogenies of *Rojas et al., 2016* was grafted onto the base tree and, after rate smoothing using the *chronopl* routine in ape *Paradis et al., 2004*, used as input in mulTree analyses. Each regression ran for 20M generations, sampling every 1000 generations, with a burn-in of 100,000. Each regression ran two separate chains, assessing post-burnin convergence by comparing posteriors and reaching estimated sampling sizes (ESS) of at least 200 for all model parameters.

Data on the dietary ecology of the species sampled were obtained from a recent study of trophic ecology (*Rojas et al., 2018*) for noctilionoids. Non noctilionoids in the subfamily Yangochiroptera are predominantly insectivorous and were coded as such. To evaluate the influence of roosting ecology on the presence of an *OPN1SW* ORF, mRNA, or S-cone, we used data from two recent studies (*Garbino and Tavares, 2018*, *Voss et al., 2016*), in addition to unpublished data for neotropical species made available by Danny Rojas.

MCMCglmm uses inverse Wishart distributions for priors on sample-wide variance and cluster-specific or phylogenetic variance. For uncorrelated predictor variables, these functions collapse into inverse gamma distributions for residuals. The choice of the residual prior is particularly important for phylogenetic logistic regressions (*Ives and Garland, 2014*), for which this variance is not identified in the likelihood (*Hadfield, 2010*). A problem arises when estimating residual variance from binary data such as presence/absence with a single trial (in this case, most species have only one observation). While there may be heterogeneity in the underlying probability of the trial, this heterogeneity is unobserved and cannot be estimated with one trial only. For this reason, many logistic regression implementations set the residual variance at zero, but this is an arbitrary choice (*Hadfield, 2016*). Instead, here we use an approach to the prior variances validated by previous comparisons of Bayesian logistic regressions (*Yohe and Dávalos, 2018*). Briefly, we set the residual variance at one (1), and allow for a flexible prior on the variance of the phylogenetic effect. In MCMCglmm notation, this prior is given by:

$$list(R = list(v = 1 fix = 1), G = list(G1 = list(V = 1, nu = 1000, alpha.mu = 0, alpha.V = 1)))$$

As the phylogenetic structure of the data corresponds to a matrix structure of correlations between observations, it becomes necessary to expand the parameters by specifying both the mean and covariance matrix on the prior, in addition to the shape and scale parameters given by nu/2. For the logistic regressions, the prior on the phylogenetic structure was given by nu = 1000 and $V$ = 1 (generating a very long-tailed distribution), with prior mean of 0 and covariance of 1.

Unlike most standard applications of MCMCglmm, our analyses of binary responses (i.e., logistic regressions of *OPN1SW* ORF, mRNA, or S-cone on various predictors) had a fixed residual variance. This produces errors in mulTree, which uses ESS and potential scale

reduction factors to determine convergence. Therefore, we modified the mulTree_fun.R R script in mulTree to overlook residual variation in its evaluation. We include this modified function in DRYAD.

The prior above, however, does not account for boundaries on coefficients. These become necessary when there is complete separation in the response variable. In our analyses, no fruit-eating species lacked mRNA, and no species roosting outside caves lacked mRNA. This complete separation yielded high coefficients for both predictors, but with very high standard errors, generating non-convergent Bayesian chains, as there is no way to calibrate the model. For this reason, we placed prior boundaries on the coefficients by adding another element to the prior for these two regressions. This element centered the prior coefficient on zero (no effect), with a variance of nine or three standard deviations. Hence the prior in these two cases was given by:

$$list\,(R = list(v = 1, fix = 1), G = list(G1 = list(V = 1, nu = 1000, alpha.mu =, alpha.V = 1)),$$
$$B = list(mu = rep(0, 2), V = diag(9, 2)))$$

In contrast to binary responses, it is feasible to estimate the residual variance of normally distributed responses such as density. Hence, for Gaussian regressions, the prior on the residuals was given by nu = 1 and $V$ = 1 (**Hadfield, 2016**), or:

$$list(R = list(V = 1, nu = 1), G = list(G1 = list(v = 1, nu = 1000, alpha.mu = 0, alpha.V = 1)))$$

Finally, we modified a script by **Smith et al. (2016)** to estimate the variance explained by the sample-wide (or fixed) effects in the models. This is also known as the conditional $R^2$, and is reported in model tables.

## RNA quality and sequencing statistics

Integrity of extracted RNA varied across samples (RIN 4.1 to 10), the majority of samples obtained RINs greater than the recommended value of 8. Following sequencing, the number of raw reads ranged from 13,221.073 to 47,639,831 per sample. Cleaned reads ranged from 12,682,257 to 45,280,391 per sample, which resulted in assemblies of 67,459 to 131,925 across species. For several species the *OPN1SW* transcript was recovered following chimera removal as the transcript was initially joined to that of *CALU*.

## Immunohistochemistry

We used IHC to assess the presence or absence of OPN1SW and OPN1LW proteins in whole, flat-mounted retinas of adult bats (**Figure 2** and **Figure 1—figure supplements 1** and **2**). We detected OPN1LW proteins in the adult eyes of all sampled bat species (*n* = 56 species total), whereas we detected the OPN1SW protein in the eyes of only 32 bat species. The following species lack the OPN1SW protein: *Molossus molossus, Eptesicus fuscus, Chilonatalus micropus, Pteronotus davyi, P. parnellii, Mormoops blainvillei, Macrotus waterhousii, Diaemus youngi, Desmodus rotundus, Trachops cirrhosus, Tonatia saurophila, Gardnerycteris crenulatum, Monophyllus redmani, Erophylla sezekorni, E. bombifrons, Brachyphylla pumila, Lonchophylla robusta*, and *Carollia brevicauda*. Specimens assayed for *Phyllostomus hastatus, Sturnira tildae, S. ludovici,* and *Platyrrhinus dorsalis*, were obtained from museum specimens. While we detected the OPN1LW protein in these samples, they were characterized by a low signal-to-background ratio in the OPN1SW protein labeling, which prevented us from determining OPN1SW presence or absence.

In most species examined either both the *OPN1SW* transcript and protein were detected (*n* = 32), or neither were detected (*n* = 5). For example, all members of the Stenodermatinae clade of fruit-eating bats examined were found to have both the *OPN1SW* transcript and protein. Outside of this clade, we also detected the OPN1SW protein, and in most cases confirmed transcript presence with RNA-Seq, in species distributed widely throughout the phylogeny. Including within Emballonuridae (*Saccopteryx bilineata* and *S. leptura*), Noctilionidae (*Noctilio leporinus*), Mormoopidae (*Pteronotus quadridens*), and other

Phyllostomidae (*Phyllostomus elongatus, Anoura geoffroyi, Glossophaga soricina, Carollia perspicillata, Rhinophylla fischerae,* and *R. pumilio*).

We detected the *OPN1SW* transcript, but no protein, in fewer species. These species are also widely distributed throughout the phylogeny and include the following: *Molossus molossus, Pteronotus parnellii, Tonatia saurophila, Gardnerycteris crenulatum, Monophyllus redmani, Erophylla bombifrons,* and *Carollia brevicauda*.

## Opsin gene evolution

The alignments used for input for the PAML analyses resulted in 349 codons for *OPN1SW*, 364 for *OPN1LW,* and 348 for *RHO*. For the analysis testing the difference in $\omega$ rates between species that do and do not express the S-cone protein, there was no difference in $\omega$ estimates between branch classes for the *RHO* ($\chi^2_{(1)}$=0.34, p=0.56; $\chi^2_{(2)}$=0.63, p=0.73). However, there was a difference favoring the three-branch class model for *OPN1SW* ($\omega_{background}$ = 0.13; $\omega_{OPN1SW.intact}$=0.24; $\omega_{OPN1SW.pseudo}$=0.78; $\chi^2_{(2)}$=70.99, p=3.84e-16) and *OPN1LW* ($\omega_{background}$ = 0.08; $\omega_{OPN1SW.intact}$=0.08; $\omega_{OPN1SW.pseudo}$=0.19; $\chi^2_{(2)}$=9.18, p=0.01) gene (Table S2 in *Supplementary file 2*). For the analysis testing for difference in $\omega$ rates between frugivorous species and background branches, there was no difference in $\omega$ estimates between branch classes for the *OPN1SW* ($\chi^2_{(1)}$=0.88, p=0.35) and *OPN1LW* ($\chi^2_{(1)}$=0.38, p=0.54) genes (Table S3 in *Supplementary file 2*). However, there was a difference in $\omega$ for the *RHO* gene (Table S3 in *Supplementary file 2*), in frugivorous lineages showed significantly lower rates than background branches ($\omega_{background}$ = 0.04; $\omega_{fugivory}$= 0.01; $\chi^2_{(1)}$=13.770.0, p=2.07e-4).

## Ecological correlates of cone presence and density

Tables S5-S8 and *Figure 4* summarize the results from Bayesian regressions. The frequency distributions of the coefficients on ecological covariates for 12 phylogenetic regressions of *OPN1SW* ORF, mRNA, or S-cone presence against diet prevalence or cave roosting reveal the prevalence of fruit as a predictor of cone presence is the strongest covariate of any model (*Figure 4*). The posterior estimates of parameters for various models model reveal the prevalence of fruit in diet increases the odds of having the S-cone by a factor of 39 (given by $e^{\hat{}}$coefficient of fruit prevalence, or ~39x higher odds of having S-cones for lineages that eat fruit than those that do not, Table S7 in *Supplementary file 2*). Analyses of the density of long-wave cones revealed no statistically meaningful effect of fruit prevalence in diet on this density (Table S8 in *Supplementary file 2*). Instead, analyses of the density of long-wave cones as a function of the presence of S-cones revealed having the S-cones increases the density of L-cones by 0.43 in the natural logarithm scale (or a factor of ~1.54 in the linear scale) compared to species without S-cones (or from a baseline of ~3944 to~6063, Table S8 in *Supplementary file 2*).

