## [Decision Letter]

[Editors’ note: this article was originally rejected after discussions between the reviewers, but the authors were invited to resubmit after an appeal against the decision.]

Thank you for submitting your work entitled "Evidence for multifactorial processes underlying phenotypic variation in bat visual opsins" for consideration by *eLife*. Your article has been reviewed by Senior Editor, a Reviewing Editor, and two reviewers.

The first reviewer indicated the manuscript failed to provide sufficient new insight into relevant aspects of evolutionary biology. This reviewer also thought that the overall claim of association between S-opsin presence and frugivory is of interest but is overly speculative. The second reviewer criticised the extent of gene and protein expression data sets required to make these claims and questioned the validity of the ecological correlation. In the light of these comments we have decided to decline the submission.

*Reviewer #1:*

A frame disruption represents only one mutation that disables the functional consequence of a protein-coding gene. Other mutations, including promoter and regulatory element mutations, will be equally effective in disabling the gene. Sadier et al. use RNA-sequencing and immunohistochemistry to study the mRNA and protein products of two cone opsin genes (OPN1SW and OPN1LW) and rhodopsin among 38 noctilionoid bats. OPN1SW disruptions have been known within the suborder Yinpterochiroptera for some time. From Figure 4, DNA sequence predicts OPN1SW protein absence 6 times, RNA transcript predicts protein absence 5 times, and that for 5 species neither DNA sequence nor RNA transcript predicted protein absence. These 5 instances (*Molossus molossus, Tonatia saurophila, Gardnerycteris crenulatum, Monophyllus redmani* and *Pteronotus parnellii*) and a sixth, the discordance between DNA or RNA and protein for *P. poeyi*, form the basis to the authors' claim in the final sentence of the Abstract. As expected, the authors show significant relaxation of constraint for lineages in which OPN1SW has become a pseudogene. Data shown in Figure 5 are proposed to indicate that S-opsin presence is partially predictive of whether a bat species is frugivorous.

The authors have clearly provided new evidence that independent OPN1SW gene loss has occurred more widely in bats, outside of the Yinpterochiroptera. Such independent losses are, however, already known to occur throughout the mammals. They also make a valid point that their "findings reveal assessments of visual perception based purely on genotypic analyses of opsin sequences, or RNA transcripts, can be misleading, and may even obscure the evolutionary processes and ecological agents of selection". They argue that variation in S-cone presence can be explained, in part, by fruit consumption. It was unclear to me how many similar hypotheses (beyond plant or flower-visiting or insectivory) that had been considered in statistical tests by the authors, and thus whether a multiple testing correction needed to have been employed.

Comparing against the aims of the journal, I came to a conclusion that this manuscript did not provide a substantially new insight into this aspect of evolutionary biology. The speculative association between S-opsin presence and frugivory is of interest, but it will remain unclear whether other (multiple?) traits could equally well explain the authors' findings.

Abstract "maintenance [of S-opsin] relates to frugivory". Clarify that this correlation is not necessarily reflective of causality.

Abstract "Discordance between DNA, RNA, and protein". In essence there is no discordance at the level of these molecules, only in our interpretation of their sequences.

Introduction "we identify specific and diverse molecular mechanisms by which selection has acted". Really?

Subsection “Variation in opsin transcripts across taxa” "Finally, we detected multiple introns in the assembled transcripts of the sister taxa *Erophylla bombifrons* and *Phyllonycteris poeyi*, suggesting transcriptional readthrough." Transcriptional readthrough is due to aberrant polyadenylation of transcripts rather than splicing out of introns, so I did not understand this statement. From the two references provided (Gaidatzis et al., 2015; Vilborg andand Steitz, 2017), I think intergenic and intronic transcription are being confused. If there is intron retention, then is there evidence that this is due to mutations at the splice site consensus sequences?

Merge Figure 1, Figure 2 and Figure 4. OPN1LW and RHO data can be presented elsewhere, for example in a Supplemental Table or Figure.

The Discussion section is over-long and could be condensed in length.

Formally, the presence of protein is not always indicative of functionality. This should be stated.

*Reviewer #2:*

In the paper the authors set out to provide "evidence for multifactorial processes underlying phenotypic variation in bat visual opsins". More specifically focusing on the correlation between DNA, RNA and protein levels of opsin genes in the noctilionoid bats. Lastly, they link such changes to diverse feeding ecologies.

It is indeed known and accepted that the relationship between the expression level of transcripts and that of the downstream proteins is not fully understood and that studies so far have shown to vary across tissues and cell types. It is indeed an important question.

My main problem with the study presented here is that the authors have not provided any tangible evidence to actually understand further the comparison between sequences, gene expression and protein levels.

Transcript presence and gene expression:

There is no analysis of gene expression, just an analysis of transcript presence (supported by the Trinity assemblies). Of course, it would not be possible to analyse gene expression when only few of the species transcriptomes (3 out of 38 if I understand correctly) have been carried out in replicates. In addition, the lack of replicate data does hamper the extant transcript analysis as well. Furthermore, the integrity of some RNA used was RIN<7 (the lowest being 4.1) – if this is the case for some samples with the addition of the lack of replicates, this is a great concern. The lower end of reads per sample was 8.6 million – again raising concerns on power.

Protein Levels: I also find the use of the Histochemistry to assess the protein level correlation to transcript very weak. I would have like to see a mass spec/targeted proteomic approach and more rigorous analysis underpinning the correlation between transcript abundance (not quantified) and protein levels.

Ecological correlation: Again, this analysis is not possible with the current datasets and methodology. Number of replicates and their usage is highly confusing. Out of 14 species there were biological replicates for 11 however, in their table S4, three of those do not appear to have biological replicates (*M. waterhousii, G. crenulatum* and *P. falcatus*), also they mention two replicates for *P. falcatus* (n=2) transcriptomes but in Table S4 I only see one.

There could be a bias towards a fruit diet based on the number of species (and their diet) used for measuring correlation; 14 species used, made up of 33 bats total – some of these had double (2 species) or triple (1 species) diets. In total, 18 individuals had a fruit/partial fruit diet, 15 insect, 3 vertebrate, and 11 plants – could the correlation biased towards fruit? there are no details on normalisation.

Museum samples: I could not understand to what extent the museum samples were used to infer the above correlations. Some clarity is necessary.

[Editors’ note: what now follows is the decision letter after the authors submitted for further consideration.]

Thank you for submitting your article "Evidence for multifactorial processes underlying phenotypic variation in bat visual opsins" for consideration by *eLife*. Your article has been reviewed by Patricia Wittkopp as the Senior Editor, a Reviewing Editor, and two reviewers. The following individual involved in review of your submission has agreed to reveal his identity: Todd Oakley (Reviewer #4).

The reviewers have discussed the reviews with one another and the Reviewing Editor has drafted this decision to help you prepare a revised submission.

Summary:

This is an interesting, and timely, investigation of opsin expression in a diverse group of bats. It provides a more in-depth study of underlying mechanisms mediating Sws1 opsin function, and loss, across a large group of bats, than many other investigations to date. Moreover, it is an unusual study in that it combines not only gene and transcriptome sequencing, with immunohistochemistry, but that it uses both freshly sampled tissue as well as museum specimens. This gives their investigation an impressive sampling across a large and extremely diverse group of bats, which has allowed them to investigate interesting questions such as the association between diet and visual abilities.

Formal reviews of this manuscript were provided by two experts in the field, but I also consulted two additional experts informally. All four of these scholars were impressed by the scope of the dataset and agreed that the work addressed an important problem. They were split, however, on how well the novelty of the findings comes through in the manuscript. In preparing a revised version, please pay special attention to this issue. Suggestions for ways to consider modifying the manuscript to address this concern are provided below. Closer integration with the literature (including some recent unpublished studies) is expected to help address this point.

In addition, the second major issue raised by the reviewers dealt with the reader's ability to assess the reliability of the data, as detailed below. Related to this the authors should think critically – and write clearly – about absence of evidence versus evidence of absence, and what that means for their overall conclusions. Basically, their main conclusions relate to absence of expression – in what cases can you be confident in your inferences; and if you cannot be 100% certain, how does that impact your conclusions?

Essential revisions:

1) Immunohistochemistry on museum specimens. Some background and information is needed, in order to better assess their results. For example, how were these museum samples fixed and stored? For how long? References to other studies that have performed similar experiments successfully are required, in order to provide context for this study, and to assess the likelihood of degradation due to storage methods.

The authors mention that they compared museum specimens with fresh sample for a few species. "For immunohistochemistry, five bat species had replicates that were both wild-caught and from museum collections and exhibited the same phenotype, highlighting the robustness of the experiments." This is important data that needs to be included in the manuscript, with figures in the supplement. Is there any evidence of degradation in older samples? On a related issue, are some of the mismatches between DNA-mRNA-protein data as shown in Figure S3 attributable to degraded museum specimens? How was a negative result determined (i.e. "Assay -ve"), as opposed to a "failure" of the assay?

2) Antibody information. Despite a central role in the immunohistochemistry experiments, almost no information is provided concerning the antibodies used to detect the S- and L-cones. Is anything known about the antibody epitopes? In which species have they been used successfully in other published studies? Is it possible that protein sequence variation resulted in a lack of S-cone antibody binding in some of the cases? Background, and references to other studies in which these antibodies were used are necessary in order to provide context, and to assess how robust antibody detection might be across different species. Citing the original papers in which the antibodies were created, and the methods used to create them are necessary. Also providing information concerning how these antibodies were obtained (from a company, from another laboratory, etc.) is essential.

3) Redundancy of the data in the figures. The data presented in Figure 1, Figure 2 and Figure 4 are redundant. The same data for OPN1SW DNA, mRNA and protein is shown twice, and the information for diet is shown three times. I agree that these three figures should be collapsed to one, to avoid this redundancy.

4) Lack of caveats and alternate interpretations of their data. In general, there is not enough discussion of caveats and limitations of their experimental approaches, nor possible alternate interpretations of their data. A few examples of this follow.

Discussion section. "In the case of opsin 1, short wave sensitive (OPN1SW) gene, while the presence (or absence) of the transcript and protein was consistent across most species, there were also multiple exceptions." This is simply a statement of the results. More interpretation of the results needs to be provided, along with caveats and limitations.

Figure 6C. Intact ORF, but no mRNA nor protein detected. Was there any evidence for mutations in regulatory regions? Figure 6D. Intact ORF and mRNA but no protein detected. Is there any evidence for protein degradation if it is a museum specimen, or sequence variation leading to substitutions in the antibody epitope region? Any evidence for abnormal post-translational modifications of the protein, leading to its degradation within the photoreceptor? Some discussion of these possible alternate interpretations is warranted.

5) Update references. This is an area of extremely active research, with several studies of bat Opn1SW published this year that were not cited. These would include Kries et al., 2018 and Gutierrez et al., 2018, which expanded opsin sampling across Neotropical lineages, and Li et al., 2018, which provides the first in vitro experimental evidence of UV sensitivity in bat Sws1 opsin. Interpreting the results presented in the MS in the context of recently published papers would help to increase the significance of this study.

6) Although the dataset is impressive and represents a major accomplishment, my major concern is the importance of the results is not clearly articulated for the broad interdisciplinary readership of *eLife*. My critique is based on my own reading but concurs with critiques of two previous reviewers.

As an attempt at synthesis, I do see the authors articulate novelty mainly along two different lines: (1) they show transcripts may not always yield protein and (2) frugivory predicts presence of SW-sensitivity in bats. At some points in the manuscript, the authors argue for novelty of their study because they are studying a new group of vertebrates (they study recently diverged mammals, whereas others studied fishes or reptiles, or anciently diverged mammals). I find the last claim (new taxon) to be uncompelling because it does not articulate what general feature about evolution they learn.

Importantly I believe the authors did not well-synthesize their first two claims for generality. First, they make a methodological critique: In order to understand links between phenotype and genotype, we must study both transcripts and proteins. This story is not synthesized with their second claim, that bat frugivory predicts color vision. Because of lack of synthesis, and because we already know that transcripts don't always lead to proteins, I'm left feeling that the authors have not spent enough time to distill and communicate clearly their results.

Still, I could imagine some general syntheses, but these would require massive re-writing, synthesis, and distillation. Alternatively, a more discipline-specific journal might also be better. If the authors do believe the best story is a methodological warning that transcripts don't predict protein – they would need to show how ignoring protein data leads to wrong conclusions. For example, what if by ignoring their protein data, they arrive at a different conclusion – maybe using transcript data alone leads to a failure to find the correlation between frugivory and color vision? If so, this provides a clear example and cautionary tale of why we cannot simply assume the protein is there – at least for cases when loss of expression is involved. It would also synthesize the main general claims.

However, my own feeling is that methodological themes are usually less broadly interesting than learning something general about biology. So, what is general here? Well, the best I can come up with along those lines is this might be an example of convergent molecular pathways leading to convergent trait loss. The general evolutionary question is – are convergent losses underpinned by the same or different genetic mechanisms? They have some hints that different changes (stop codons, intron read-through, etc.) might interrupt opsin expression in different species (I didn't look carefully at the distribution of losses on the tree though). A paper that does a good job of setting up the general question of parallel loss (although morphology) is (Sumner-Rooney et al. 2016). A similar topic is also studied for flower color (Zufall and Rausher, 2004). Although parallel loss is fairly well studied, I don't think different failures of translation are known in parallel. I do see two main challenges to taking this approach though. (1) the authors will not know which mutations caused the failure to express proteins versus those that came later (2) I'd like to be convinced that the sequences found with RNA-seq can be replicated with PCR because transcriptome assembly can be error-prone.

Of course, the authors know the data far better than I do and so I am not really trying to dictate what the synthesis is (also my suggestions don't incorporate their dn/ds analyses). Rather, I provided two examples to try to articulate more clearly what I feel is lacking from the current manuscript.

This is a very valuable data set in its breadth. It is indeed rare to have transcripts and protein expression in broad comparisons. However, current descriptions of the importance of the work go in multiple directions, in my opinion leaving any one direction insufficient. I am open to the possibility that the writing (namely the synthesis and communication of the importance; sometimes called "novelty") could be improved, providing a story of very general interest across disciplines for *eLife*. But this would take a rather major re-write.

References

Sumner-Rooney L., Sigwart J.D., McAfee J., Smith L., Williams S.T. 2016. Repeated eye reduction events reveal multiple pathways to degeneration in a family of marine snails. Evolution. 70:2268-2295.

Zufall R.A., Rausher M.D. 2004. Genetic changes associated with floral adaptation restrict future evolutionary potential. Nature. 428:847-850.

---

## [Author Response]

[Editors’ note: the author responses to the first round of peer review follow.]

The first reviewer indicated the manuscript failed to provide sufficient new insight into relevant aspects of evolutionary biology. This reviewer also thought that the overall claim of association between S-opsin presence and frugivory is of interest but is overly speculative. The second reviewer criticised the extent of gene and protein expression data sets required to make these claims and questioned the validity of the ecological correlation. In the light of these comments we have decided to decline the submission.

Thank you for the opportunity to provide a rebuttal of the reviews on our paper ‘Evidence for multifactorial processes underlying phenotypic variation in bat visual opsins’. Between them the two reviewers raised three issues: (1) the novelty of the findings, (2) the extent of sampling and (3) the ecological models. We address these in turn below.

1) Novelty of findings

To our knowledge, while there have been numerous studies of opsin gene evolution in mammals, there has not yet been a single study that has attempted to relate coding sequences to the presence of transcripts and proteins in a comparative framework. Moreover, almost all studies of mammals have focused at deeper evolutionary timeframes. By revealing that opsin phenotypes are highly variable among even closely related bat species (sometimes even between congeners) within a single monophyletic clade, and that this variation stems from diverse molecular routes involving each step of the central dogma, we feel we are making a significant contribution to our understanding of the evolution of color vision across mammals.

Furthermore, it has not previously been shown that mammalian species that possess seemingly intact opsin gene sequences may still lack the corresponding cone proteins. Therefore, this finding suggests we may be underestimating phenotypic variation in mammalian color vision across species, which could have wider implications for divergent taxonomic groups such as primates.

2) The extent of sampling

Reviewer 2 raised concerns regarding the extent of sampling, in terms of biological replicates, for the gene expression and protein components of the study. While we agree it would be desirable to sample larger numbers of biological replicates to estimate variance, as is routine in studies of insects and lab animals, this was not possible due the nature of our study (i.e. on wild mammals – some rare – that required numerous ethical and research permits in order to sample). Additionally, there is a hard ceiling on the number of offspring most bats can have limiting their population recovery. For this reason, as well as the many external threats to wild populations, permitting agencies are loath to grant collecting permits for many individuals unless a population is known to be large, common, and unthreatened. We suspect that with the possible exception of some fishes, the same restrictions would apply to most other wild vertebrates. For this reason, we feel that enforcing comparable standards is not practical and would inevitably limit knowledge to restricted groups of model taxa.

Our study group of bats are arguably unparalleled among mammal in their ecological diversity within a single superfamily, and thus are highly suited for the broad comparative approach we opted to take. Increasing technical and biological replicates would have come with the cost of reduced taxonomic sampling which would ultimately have limited the ecological component of our study and thus also the significance of our findings. We are confident that our findings will have implications for other key groups that have been the focus of opsin studies (including primates, which would be even more difficult to sample for RNA).

3) Ecological models

To address these questions, we have added additional information and explanation to the text detailing these methods. The hierarchical Bayesian approach we adopted to analyse the relationship between diet and opsin phenotype is widely used in evolutionary ecological studies of trait evolution (e.g. Wagner et al., (2012); Lukas andand Clutton-Brock, (2013).

The evolutionary models we use are based on a Bayesian hierarchical format (and not a classical, frequentist approach), therefore, we are not testing a null hypothesis of no effect but instead modelling the distribution of an effect, whatever its size. Therefore, there is not the same requirements for adjustments following the analyses of multiple hypotheses, we detail this further below.

Reviewer #1:A frame disruption represents only one mutation that disables the functional consequence of a protein-coding gene. Other mutations, including promoter and regulatory element mutations, will be equally effective in disabling the gene. Sadier et al. use RNA-sequencing and immunohistochemistry to study the mRNA and protein products of two cone opsin genes (OPN1SW and OPN1LW) and rhodopsin among 38 noctilionoid bats. OPN1SW disruptions have been known within the suborder Yinpterochiroptera for some time. From Figure 4, DNA sequence predicts OPN1SW protein absence 6 times, RNA transcript predicts protein absence 5 times, and that for 5 species neither DNA sequence nor RNA transcript predicted protein absence. These 5 instances (Molossus molossus, Tonatia saurophila, Gardnerycteris crenulatum, Monophyllus redmani and Pteronotus parnellii) and a sixth, the discordance between DNA or RNA and protein for P. poeyi, form the basis to the authors' claim in the final sentence of the Abstract. As expected, the authors show significant relaxation of constraint for lineages in which OPN1SW has become a pseudogene. Data shown in Figure 5 are proposed to indicate that S-opsin presence is partially predictive of whether a bat species is frugivorous.The authors have clearly provided new evidence that independent OPN1SW gene loss has occurred more widely in bats, outside of the Yinpterochiroptera. Such independent losses are, however, already known to occur throughout the mammals. They also make a valid point that their "findings reveal assessments of visual perception based purely on genotypic analyses of opsin sequences, or RNA transcripts, can be misleading, and may even obscure the evolutionary processes and ecological agents of selection".

We are pleased reviewer 1 recognises that our findings are valid. We further clarify the novelty and significance of our findings in terms of both bat and mammalian evolution below.

They argue that variation in S-cone presence can be explained, in part, by fruit consumption. It was unclear to me how many similar hypotheses (beyond plant or flower-visiting or insectivory) that had been considered in statistical tests by the authors, and thus whether a multiple testing correction needed to have been employed.

This would make sense if we were using a classical, frequentist approach, but we are using Bayesian hierarchical models. Unlike classical statistics, the Bayesian approach we use has two advantages. First, we are not testing a null hypothesis of no effect but instead modelling the distribution of an effect, whatever its size. Second, a hierarchical model shifts points estimates (i.e., the mean by group) and their confidence intervals toward each other and thus does not suffer from the stationarity of point estimates and the need to widen intervals in multiple comparisons that emerges from classical statistics. A direct quote from Gelman et al., (2012) elaborates on these points:

“First, we are typically not terribly concerned with Type 1 error because we rarely believe that it is possible for the null hypothesis to be strictly true. Second, we believe that the problem is not multiple testing but rather insufficient modeling of the relationship between the corresponding parameters of the model. Once we work within a Bayesian multilevel modeling framework and model these phenomena appropriately, we are actually able to get more reliable point estimates. A multilevel model shifts point estimates and their corresponding intervals toward each other (by a process often referred to as “shrinkage” or “partial pooling”), whereas classical procedures typically keep the point estimates stationary, adjusting for multiple comparisons by making the intervals wider (or, equivalently, adjusting the p values corresponding to intervals of fixed width).” From Gelman et al. (2012). Why We (Usually) Don't Have to Worry About Multiple Comparisons. Journal of Research on Educational Effectiveness 5(2): 189-211.”

We have also clarified how many models were applied in the Materials and methods section and Results section.

Comparing against the aims of the journal, I came to a conclusion that this manuscript did not provide a substantially new insight into this aspect of evolutionary biology. The speculative association between S-opsin presence and frugivory is of interest, but it will remain unclear whether other (multiple?) traits could equally well explain the authors' findings.

As highlighted in our paper much work has indeed been performed on assessing OPN1SW gene loss both in bats and other mammals (e.g. Emerling et al., 2015; Emerling et al., 2017; Zhao et al., 2009). Within bats, it has only previously been shown that some species possess pseudogenized OPN1SW these have all been inferred based on the presence of disrupted ORFs and the presence of STOP codons. Indeed, at the same time as our paper was submitted for review high impact studies are being published (e.g. Gutierrez et al., (2018) that are continuing to infer functional S-cones based on the presence of mRNA. Furthermore, we are able to show that several species in which OPN1SW has been inferred to be functional (e.g. *Pteronotus parnellii* as published in Emerling et al., 2015 and Gutierrez et al., 2018) is in fact not expressed as a protein.

Therefore, while it has been suggested that OPN1SW gene loss has occurred across mammals we are able to show that the prevalence of this is actually much higher than has previously been considered and is still happening (in the process of been lost?). Therefore, we are presenting data which provides a major new insight into the true patterns of short-wave sensitive vision across mammals, tnat could also be applied to many other genes and taxa. Together, these data are within the aims of the journal.

Our model systems of ecologically diverse, yet phylogenetically related, neotropical bat species is able to exemplify this pattern by highlighting the short evolutionary time periods that the diversity in colour vision has evolved. Previous accounts of S-opsins in mammals have often been linked to changes in photic environment e.g. shifts to aquatic/subterranean environments, therefore, our study also provides new insights into the potential drivers of OPN1SW gene loss as we see differences in its retention in closely related species that inhabitant identical terrestrial habitats.

Abstract "maintenance [of S-opsin] relates to frugivory". Clarify that this correlation is not necessarily reflective of causality.

This sentence has been removed from the current version, and we have reworded other sections.

Abstract "Discordance between DNA, RNA, and protein". In essence there is no discordance at the level of these molecules, only in our interpretation of their sequences.

We are unsure of what this means. There is discordance when the putatively functional sequence is present and yet its protein product is not. We now clarify what is meant in the Abstract.

Introduction "we identify specific and diverse molecular mechanisms by which selection has acted". Really?

To avoid confusion, we have rephrased the sections of the manuscript where the term ‘mechanism’ was used.

Subsection “Variation in opsin transcripts across taxa” "Finally, we detected multiple introns in the assembled transcripts of the sister taxa Erophylla bombifrons and Phyllonycteris poeyi, suggesting transcriptional readthrough." Transcriptional readthrough is due to aberrant polyadenylation of transcripts rather than splicing out of introns, so I did not understand this statement. From the two references provided (Gaidatzis et al., 2015; Vilborg and Steitz, 2017), I think intergenic and intronic transcription are being confused. If there is intron retention, then is there evidence that this is due to mutations at the splice site consensus sequences?

We thank the reviewer for highlighting our error here, we had indeed through an oversight used the incorrect term and references at this point.

Merge Figure 1, Figure 2 and Figure 4. OPN1LW and RHO data can be presented elsewhere, for example in a Supplemental Table or Figure.

We have opted to present the datasets separately in this way to may it clearer exactly which data points correspond to each particular part of the study. However, if the consensus agreement across reviewers and editors is to merge the figures, we are happy to do so.

The Discussion section is over-long and could be condensed in length.

We have shortened the Discussion section by two paragraphs.

Formally, the presence of protein is not always indicative of functionality. This should be stated.

We now state this in the results of IHC.

Reviewer #2:In the paper the authors set out to provide "evidence for multifactorial processes underlying phenotypic variation in bat visual opsins". More specifically focusing on the correlation between DNA, RNA and protein levels of opsin genes in the noctilionoid bats. Lastly, they link such changes to diverse feeding ecologies.It is indeed known and accepted that the relationship between the expression level of transcripts and that of the downstream proteins is not fully understood and that studies so far have shown to vary across tissues and cell types. It is indeed an important question.

We are pleased that the reviewer recognises that the motivation for our study is as an important issue.

My main problem with the study presented here is that the authors have not provided any tangible evidence to actually understand further the comparison between sequences, gene expression and protein levels.

Levels of opsin transcripts are known to fluctuate with circadian rhythm and exposure to light around the time of sampling, and thus assessing variation in expression levels would require biological replicates to account for sample variance. While this is desirable experimentally, it would mean euthanizing many more individuals and is thus not feasible for wild mammals, either ethically or in terms of obtaining permissions. By measuring the presence or absence of opsin transcripts we provide a more robust result given these limitations. See below for more details.

Transcript presence and gene expression:There is no analysis of gene expression, just an analysis of transcript presence (supported by the Trinity assemblies). Of course, it would not be possible to analyse gene expression when only few of the species transcriptomes (3 out of 38 if I understand correctly) have been carried out in replicates. In addition, the lack of replicate data does hamper the extant transcript analysis as well. Furthermore, the integrity of some RNA used was RIN<7 (the lowest being 4.1) – if this is the case for some samples with the addition of the lack of replicates, this is a great concern. The lower end of reads per sample was 8.6 million – again raising concerns on power.

RIN scores vary across samples due the challenging nature of sample collection, but if the quality of the RNA determined the detection of OPN1SW transcripts, then species lacking transcripts would also tend to have low RIN. Instead, we find no relationship between RIN and presence of OPN1SW. For example, the species with the lowest RIN (*Pteronotus quadridens* 4.1) was found to contain a nearly intact OPN1SW mRNA transcript. The same is true for reads per sample. Additionally, in all bats sampled we were able to recover complete or nearly complete OPN1LW and RHODOPSIN sequences regardless of variation in RIN or sequencing depth. For the three species that we do have biological replicates we have consistent results in terms of mRNA presence/absence across all three visual pigments. We can add measures of transcriptome completeness (e.g. BUSCO scores) to the paper if required.

Protein Levels: I also find the use of the Histochemistry to assess the protein level correlation to transcript very weak. I would have like to see a mass spec/targeted proteomic approach and more rigorous analysis underpinning the correlation between transcript abundance (not quantified) and protein levels.

While documenting the presence or absence of either transcript or protein implies some level of quantitative correspondence, correlating the levels of mRNA and protein is not within the scope of this study. We did not attempt to estimate protein level – we primarily aimed at documenting presence or absence of OPN1SW and OPN1LW. Given the contrasting methods necessary for preserving either mRNA or protein (RNAlater vs. 4% paraformaldehyde), and as it was only possible to assay protein and not mRNA from museum specimens, we did not use the same individual for the RNA-Seq and IHC studies. As different individuals were used for RNA/protein detection assessing correlations in terms of mRNA expression/protein level are beyond the scope of the study. Furthermore, a number of studies suggest that there should not be a direct correlation between mRNA and protein level, for example, sometimes mRNA levels regulate gene expression etc.

Our estimates of S/L-cone density measures the distribution of the cones across the retina, so they correspond to protein levels only to the extent that having more cones indicates having more protein. For our purposes, density is more informative of how the distribution varies across the surface. A simple quantification in terms of protein level may be difficult to adjust given the size differences of the eyes.Ecological correlation: Again, this analysis is not possible with the current datasets and methodology. Number of replicates and their usage is highly confusing. Out of 14 species there were biological replicates for 11 however, in their table S4, three of those do not appear to have biological replicates (M. waterhousii, G. crenulatum and P. falcatus), also they mention two replicates for P. falcatus (n=2) transcriptomes but in Table S4 I only see one.There could be a bias towards a fruit diet based on the number of species (and their diet) used for measuring correlation; 14 species used, made up of 33 bats total – some of these had double (2 species) or triple (1 species) diets. In total, 18 individuals had a fruit/partial fruit diet, 15 insect, 3 vertebrate, and 11 plants – could the correlation biased towards fruit? there are no details on normalisation.

We have clarified the language in the text and the tables so it is clear when there are and there aren't biological replicates. Two kinds of regressions were conducted. The first one related the presence of cones to the diet of species. In this case each species was represented by one observation. There were 49 species, 29 of which included fruit in their diet.

The other type of regression modelled the density from 1-several individuals within species. In this case, we take advantage of the hierarchical structure of the data in which observations (individuals) cluster within species. The variance within and between species is both included in the model and modelled separately. Hence the model simultaneously captures variance between replicates (when present) and between species, when estimating the means corresponding to the groups of interest. We now indicate the natural log transformation, which served to normalize the density data.

For the 14 bat species that were used for the ecological correlates of opsin density models we state in the Material and methods section that “The accuracy of this automatic approach was verified by manually counting three biological replicates of five bat species, by two different people.”

Specifically, regarding the confusion surrounding the *P. falcatus* samples, two individuals were sampled for the transcriptome analyses (DR171 and DR003, as detailed in the RNAseq data tab of Supplementary table S1). Whereas, Table S4 presents the cone densities in the 14 species representative of different diet types and is based on IHC data and inferred dietary data only.

Museum samples: I could not understand to what extent the museum samples were used to infer the above correlations. Some clarity is necessary.

We have provided complete sample information in Supplementary file 1 and Figure 1—Figure Supplement 1, to clearly state which samples were wild caught and which were museum samples.

[Editors’ note: the author responses to the re-review follow.]

Summary:This is an interesting, and timely, investigation of opsin expression in a diverse group of bats. It provides a more in-depth study of underlying mechanisms mediating Sws1 opsin function, and loss, across a large group of bats, than many other investigations to date. Moreover, it is an unusual study in that it combines not only gene and transcriptome sequencing, with immunohistochemistry, but that it uses both freshly sampled tissue as well as museum specimens. This gives their investigation an impressive sampling across a large and extremely diverse group of bats, which has allowed them to investigate interesting questions such as the association between diet and visual abilities.Formal reviews of this manuscript were provided by two experts in the field, but I also consulted two additional experts informally. All four of these scholars were impressed by the scope of the dataset and agreed that the work addressed an important problem. They were split, however, on how well the novelty of the findings comes through in the manuscript. In preparing a revised version, please pay special attention to this issue. Suggestions for ways to consider modifying the manuscript to address this concern are provided below. Closer integration with the literature (including some recent unpublished studies) is expected to help address this point.In addition, the second major issue raised by the reviewers dealt with the reader's ability to assess the reliability of the data, as detailed below. Related to this the authors should think critically – and write clearly – about absence of evidence versus evidence of absence, and what that means for their overall conclusions. Basically, their main conclusions relate to absence of expression – in what cases can you be confident in your inferences; and if you cannot be 100% certain, how does that impact your conclusions?

We thank the Reviewing Editor and the four reviewers for these constructive comments concerning our manuscript. We have made extensive revisions to our study following these recommendations, these include the addition of new figures to more clearly display our data and protocol, and considerable restructuring of the focus of the study to make it more accessible to readers from a wider range of subject. Please see our detailed response to the specific comments raised below.

Essential revisions:1) Immunohistochemistry on museum specimens. Some background and information is needed, in order to better assess their results. For example, how were these museum samples fixed and stored? For how long? References to other studies that have performed similar experiments successfully are required, in order to provide context for this study, and to assess the likelihood of degradation due to storage methods.

We agree that more information regarding the museum specimens was needed in our paper. We have now added the following references: Hühns et al., 2015 and Nießner et al., 2016. The former describes the utility of museum samples for a range of molecular approaches, and the latter presents information on the use of the same S-opsin antibody across a range of mammal species, preserved by a range of methods. In addition, we added the collection date of the museum and field specimens used for the IHC assays to supplementary table S1, we also present data in Figure 1—figure supplement 4 to show that the staining is consistent across museum specimens (including the oldest specimen collected in 1921), and the IHC protocol is published in Ortín-Martínez et al., 2015, which we cite.

Museum samples were originally fixed in 70% ethanol and stored at room temperature. After dissecting out tissues from museum samples, we re-fixed in PFA and processed. All of these details have now been added to the manuscript.

The authors mention that they compared museum specimens with fresh sample for a few species. "For immunohistochemistry, five bat species had replicates that were both wild-caught and from museum collections and exhibited the same phenotype, highlighting the robustness of the experiments." This is important data that needs to be included in the manuscript, with figures in the supplement. Is there any evidence of degradation in older samples? On a related issue, are some of the mismatches between DNA-mRNA-protein data as shown in Figure S3 attributable to degraded museum specimens? How was a negative result determined (i.e. "Assay -ve"), as opposed to a "failure" of the assay?

We have added supplementary Figure 1—figure supplement 4 to display representative staining of both field caught and museum specimens from the same species and therefore, compare the consistency of the opsins detection.

We have added information regarding degradation of older samples to the manuscript. Basically, the oldest specimen we looked at had solid L- and S- cone staining so age in and of itself was not an issue. We did have instances in which we could detect L- but not S- cones however, that were probably due to preservation issues. To distinguish between real and false (non-)signals, we developed a stringent set of criteria based on tissue preservation state, and also completed staining on multiple samples and individuals whenever possible. We also now address the relationship between mismatches and the possible preservation/methodological issues explicitly in the manuscript. Given our sample sizes, methods, and tissue evaluation, we are confident that any non-signals (for species included in our analyses) represent real non-signals. However, we also explicitly discuss the possible issues with our approach and alternative interpretations in the manuscript.

2) Antibody information. Despite a central role in the immunohistochemistry experiments, almost no information is provided concerning the antibodies used to detect the S- and L-cones. Is anything known about the antibody epitopes? In which species have they been used successfully in other published studies? Is it possible that protein sequence variation resulted in a lack of S-cone antibody binding in some of the cases? Background, and references to other studies in which these antibodies were used are necessary in order to provide context, and to assess how robust antibody detection might be across different species. Citing the original papers in which the antibodies were created, and the methods used to create them are necessary. Also providing information concerning how these antibodies were obtained (from a company, from another laboratory, etc.) is essential.

The following primary antibodies were used: goat anti-OPN1SW, 1∶1000 (sc-14363, Santa-Cruz Biotechnologies, Heidelberg, Germany; detects S-opsin protein) and rabbit anti-opsin red/green, 1∶750 (ab5405, Millipore Ibérica, Madrid, Spain; detects L-opsin protein). AB5405 was raised in rabbit against the last 42 amino acids at the C-terminus of recombinant human red/green opsins. Sc-14363 is an affinity-purified goat polyclonal antibody raised against a 20-amino-acid synthetic peptide mapping within amino acids 1 to 50 of human blue-sensitive opsin. The antibodies were designed by commercial companies, and not for a specific study or manuscript. Both of these antibodies have been used successfully in many species, including but not limited to bats, rodents, artiodactyls, etc. This information has now been added to the manuscript. We also provide figures of the amino acid alignments for all the bats included in our study for which we have sequence data for in these regions so that the degree of conservation can be accessed by readers. Based on these alignments we are confident that lack of the S-opsin binding in the species which gave a negative result are not due to increased sequence variation.

3) Redundancy of the data in the figures. The data presented in Figure 1, Figure 2 and Figure 4 are redundant. The same data for OPN1SW DNA, mRNA and protein is shown twice, and the information for diet is shown three times. I agree that these three figures should be collapsed to one, to avoid this redundancy.

As suggested, we now present this data as a single figure.

4) Lack of caveats and alternate interpretations of their data. In general, there is not enough discussion of caveats and limitations of their experimental approaches, nor possible alternate interpretations of their data. A few examples of this follow.Discussion section. "In the case of opsin 1, short wave sensitive (OPN1SW) gene, while the presence (or absence) of the transcript and protein was consistent across most species, there were also multiple exceptions." This is simply a statement of the results. More interpretation of the results needs to be provided, along with caveats and limitations.

We agree with the reviewers and have now included a new subsection “Study limitations and alternative interpretations”. In this section we have discussed whether some of our findings might have arisen as artifacts due to problems with museum specimens, data assembly, sample sizes and cryptic diversity. Although we acknowledge there are a small number of cases where artifactual results cannot be completely ruled out, many of the results are supported by other recent findings, and the overarching finding (of multiple routes and steps underpinning parallel degradation) does not change. In this section we have now included additional references to papers that support our data and interpretations.

Figure 6C. Intact ORF, but no mRNA nor protein detected. Was there any evidence for mutations in regulatory regions? Figure 6D. Intact ORF and mRNA but no protein detected. Is there any evidence for protein degradation if it is a museum specimen, or sequence variation leading to substitutions in the antibody epitope region? Any evidence for abnormal post-translational modifications of the protein, leading to its degradation within the photoreceptor? Some discussion of these possible alternate interpretations is warranted.

For the two species with putatively intact ORFs but with no mRNA or protein, genomic resources are currently unavailable so are not yet able to study the regulatory regions. This information has been added to the manuscript. For the cases where we find mRNA and no protein, we have now included additional evidence that suggests that in some species the ORF is in fact disrupted at the level of mRNA. We have also included information relating to the origin of the samples used for the protein assay and sequence conservation of the protein in the area of the epitope (see new Figure 1—figure supplement 2 and Figure 1—figure supplement 3). We agree that we needed to explain our quality control measures that we used to ensure the quality of our retinas before staining. We added a section in the Material and methods section which describe the different steps of our quality control to keep or discard a retina from our dataset. While protein degradation was not proven per se, we carefully inspected the anatomy of the retina to ensure its integrity before staining. In addition, we checked the morphology of the cones after staining. We have not found any evidence of mutation leading to protein degradation within the photoreceptor.

5) Update references. This is an area of extremely active research, with several studies of bat Opn1SW published this year that were not cited. These would include Kries et al., 2018 and Gutierrez et al., 2018, which expanded opsin sampling across Neotropical lineages, and Li et al., 2018, which provides the first in vitro experimental evidence of UV sensitivity in bat Sws1 opsin. Interpreting the results presented in the MS in the context of recently published papers would help to increase the significance of this study.

We thank the reviewers for highlighting these studies that have been published since we initially submitted our paper. These studies have now been referenced in our paper, and we have also been able to incorporate some of the data produced by these to confirm some of our findings, e.g. the sequences that we recovered based on transcriptome assembly match those produced by PCR in these studies (also see below). As pointed out due to the area of active research we have also included reference to Wu et al., 2018 and Simoes et al., 2018.

6) Although the dataset is impressive and represents a major accomplishment, my major concern is the importance of the results is not clearly articulated for the broad interdisciplinary readership of eLife. My critique is based on my own reading but concurs with critiques of two previous reviewers.As an attempt at synthesis, I do see the authors articulate novelty mainly along two different lines: (1) they show transcripts may not always yield protein and (2) frugivory predicts presence of SW-sensitivity in bats. At some points in the manuscript, the authors argue for novelty of their study because they are studying a new group of vertebrates (they study recently diverged mammals, whereas others studied fishes or reptiles, or anciently diverged mammals). I find the last claim (new taxon) to be uncompelling because it does not articulate what general feature about evolution they learn.Importantly I believe the authors did not well-synthesize their first two claims for generality. First, they make a methodological critique: In order to understand links between phenotype and genotype, we must study both transcripts and proteins. This story is not synthesized with their second claim, that bat frugivory predicts color vision. Because of lack of synthesis, and because we already know that transcripts don't always lead to proteins, I'm left feeling that the authors have not spent enough time to distill and communicate clearly their results.Still, I could imagine some general syntheses, but these would require massive re-writing, synthesis, and distillation. Alternatively, a more discipline-specific journal might also be better. If the authors do believe the best story is a methodological warning that transcripts don't predict protein – they would need to show how ignoring protein data leads to wrong conclusions. For example, what if by ignoring their protein data, they arrive at a different conclusion – maybe using transcript data alone leads to a failure to find the correlation between frugivory and color vision? If so, this provides a clear example and cautionary tale of why we cannot simply assume the protein is there – at least for cases when loss of expression is involved. It would also synthesize the main general claims.However, my own feeling is that methodological themes are usually less broadly interesting than learning something general about biology. So, what is general here? Well, the best I can come up with along those lines is this might be an example of convergent molecular pathways leading to convergent trait loss. The general evolutionary question is – are convergent losses underpinned by the same or different genetic mechanisms? They have some hints that different changes (stop codons, intron read-through, etc.) might interrupt opsin expression in different species (I didn't look carefully at the distribution of losses on the tree though). A paper that does a good job of setting up the general question of parallel loss (although morphology) is (Sumner-Rooney et al. 2016). A similar topic is also studied for flower color (Zufall and Rausher 2004). Although parallel loss is fairly well studied, I don't think different failures of translation are known in parallel. I do see two main challenges to taking this approach though. (1) the authors will not know which mutations caused the failure to express proteins versus those that came later (2) I'd like to be convinced that the sequences found with RNA-seq can be replicated with PCR because transcriptome assembly can be error-prone.Of course, the authors know the data far better than I do and so I am not really trying to dictate what the synthesis is (also my suggestions don't incorporate their dn/ds analyses). Rather, I provided two examples to try to articulate more clearly what I feel is lacking from the current manuscript.This is a very valuable data set in its breadth. It is indeed rare to have transcripts and protein expression in broad comparisons. However, current descriptions of the importance of the work go in multiple directions, in my opinion leaving any one direction insufficient. I am open to the possibility that the writing (namely the synthesis and communication of the importance; sometimes called "novelty") could be improved, providing a story of very general interest across disciplines for eLife. But this would take a rather major re-write.

The reviewers acknowledged the value of our dataset in terms of its breadth, but also had a number of recommendations for improving the paper’s angle to highlight the novelty of the findings. We found these comments to be particularly constructive and have taken them onboard. We hope that the reviewers will agree with us that the paper is much improved as a result.

Briefly, to make our results of wider interest, and more accessible to the readers of *eLife*, we have restructured the manuscript. Our revision contains two new analyses, references to new supporting literature, and highlights our most salient findings more clearly.

Briefly, we have reworked the paper to emphasize the parallel and independent nature of losses of OPN1SW, in line with recommendations of the reviewer (“convergent molecular pathways leading to convergent trait loss…are convergent losses underpinned by the same or different genetic mechanisms?”). To obtain further insights into the pathways of loss, for our revision we performed reconstructions of mRNA transcripts for the three genes for each taxon for which we have RNA-seq data. We then examined the integrity and diversity of mRNA transcripts in relation to the presence of the ospins proteins. The findings showed that complete RHO and OPN1LW mRNA transcripts were almost always present, whereas OPN1SW transcripts were highly variable among taxa with respect to the number of isoforms, and, in some cases, exons. We validated our findings with PCRs and other published data whenever possible and we discuss our results in the light of similar isolated reports of splice variation in ospin sequences. We show these mRNA transcripts in a new figure, Figure 4. An important finding of this analysis was the potentially high number of independent pathways of degradation.

The reviewer also made the point that we could make more of our ecological models (“if the authors do believe the best story is a methodological warning that transcripts don't predict protein – they would need to show how ignoring protein data leads to wrong conclusions”). We agree with this too and have now addressed this comment by expanding the models to each step of the synthesis of S-opsin: DNA, mRNA and protein. Although several new manuscripts have pointed out that cave roosting ecology may play a role in the evolution of parallel losses, none have formally tested this relationship using quantitative models. Therefore, we have included cave roosting as a factor and now discuss the results. Our results hold: the protein data have the strongest links to the ecology of the lineages, as evidenced by the coefficients (which are directly comparable because they all correspond to the multiplier on the presence of a given ecology). These results are now presented in Tables S5-S7 in Supplementary file 2, as well as Figure 5. Using odds ratios based on the coefficients we show that frugivory is the best predictor for the presence of mRNA and protein, and that protein presence has the strongest relationship to ecology of all the data sets. To our knowledge this is the first time a study has attempted to formally quantify these relationships across a broad comparative sample. Finally, ecological models are not mutually exclusive with a focus on parallel routes of loss. Instead, the full knowledge of the routes of loss has a bearing on the ecological model results.

To summarise, combining our new findings with our original analyses, we believe that our study is stronger and has the potential to become a textbook example of how parallel losses can shape trait evolution among closely related species. While this idea is not new, our study is important because it shows that losses of a trait (in our case OPN1SW protein synthesis in neotropical bats) can arise in parallel by multiple different genetic routes. Furthermore, consideration of each these stages must be considered in order to obtain a full picture of evolutionary loss, which would otherwise be underestimated. Parallel losses are often investigated in relatively divergent species and are never studied at all the main three levels of regulation. We are able to show that these independent losses occur through different mechanisms over evolutionary time. Finally, we reveal that the complex patterns of parallel losses occur in response to ecological pressures. We have been able to confirm via a number of sources (i.e. published RNA-Seq transcripts, PCRs and genomes) that the DNA ORFs recovered by our RNA-Seq assemblies are correct. At the same time, we acknowledge that we are unable to verify our alternate isoforms using PCR or genomic assemblies. Finally, while our findings could apply to any trait, the fact that these results come from opsin genes is likely to be of very wide interest. There have been many recent papers on opsins, published in leading journals, but nearly all studies consider genes alone, and relate observations of mutations to ecological habits without statistical support.